# Taxonomic Novelties and New Records of *Amanita* Subgenus *Amanitina* from Thailand

**DOI:** 10.3390/jof9060601

**Published:** 2023-05-24

**Authors:** Yuan S. Liu, Jian-Kui Liu, Jaturong Kumla, Nakarin Suwannarach, Saisamorn Lumyong

**Affiliations:** 1Department of Biology, Faculty of Science, Chiang Mai University, Chiang Mai 50200, Thailand; yuanshuailiu9@gmail.com (Y.S.L.); jaturong_yai@hotmail.com (J.K.); suwan.462@gmail.com (N.S.); 2Doctor of Philosophy Program in Applied Microbiology (International Program), Faculty of Science, Chiang Mai University, Chiang Mai 50200, Thailand; 3Research Center of Microbial Diversity and Sustainable Utilization, Chiang Mai University, Chiang Mai 50200, Thailand; 4School of Life Science and Technology, Center for Informational Biology, University of Electronic Science and Technology of China, Chengdu 611731, China; liujiankui@uestc.edu.cn; 5Academy of Science, The Royal Society of Thailand, Bangkok 10300, Thailand

**Keywords:** Amanitaceae, multi-gene phylogeny, mycorrhizal fungi, species diversity, three new species

## Abstract

The *Amanita* subgenus *Amanitina* contains six sections, and the species diversity of this subgenus has still not been explored in Thailand. Twenty samples collected in 2019 and 2020, which had the morphological characteristics of the *Amanita* subgen. *Amanitina*, were observed in this study. Both the microscopical characteristics and multi-gene phylogenetic analyses of the ITS, nrLSU, *RPB2*, *TEF1-α*, and *TUB* gene regions revealed that the 20 samples represented nine species and dispersed into four sections. Remarkably, three taxa were different from any other currently known species. Here, we describe them as new to science, namely *A. albifragilis*, *A. claristriata*, and *A. fulvisquamea*. Moreover, we also recognized six interesting taxa, including four records that were new to Thailand, viz. *A. cacaina*, *A. citrinoannulata*, *A. griseofarinosa*, and *A. neoovoidea*, as well as two previously recorded species, *A. caojizong* and *A. oberwinkleriana*. Moreover, we provide the first *RPB2* and *TEF1-α* gene sequences for *A. cacaina*. Detailed descriptions, illustrations as line drawings, and comparisons with related taxa are provided.

## 1. Introduction

*Amanita* Pers. is an important basidiomycetous genus comprising about 700 species [1,2,3,4]. It contains both well-known edible and deadly poisonous species. In addition, *Amanita* species are regarded as key organisms involved in nutrient and carbon cycling in forest ecosystems on account of their ability to form ectomycorrhizal relationships with more than 10 families of vascular plants, e.g., Dipterocarpaceae, Fagaceae, Myrtaceae, and Pinaceae [2,5,6,7].

Since the genus *Amanita* was formally established in 1797 [8], many mycologists have continued to contribute to and improve the taxonomic knowledge of this genus [9,10,11,12,13,14,15,16,17,18,19,20,21,22]. Corner and Bas [15] and Bas [16] proposed splitting the genus *Amanita* into two subgenera and six sections, which had important significance for the taxonomy of *Amanita*. Yang [17] revised the classification of this genus and split it into two subgenera and seven sections. Although the above classifications have been widely adopted, the delimitation within this genus is still controversial [23,24]. Until 2018, according to multi-gene phylogenetic analysis, morphological examinations, and ecological studies, Cui et al. [2] proposed the division of the genus *Amanita* into three subgenera and eleven sections as follows. The subgenus *Amanita* contains the section *Amanita*, section *Amarrendiae* (Bougher and T. Lebel) Zhu L. Yang, Y.Y. Cui, Q. Cai, and L.P. Tang, section *Caesareae* Singer ex Singer, and section *Vaginatae* (Fr.) Quél. The subgenus *Amanitina* (E. J. Gilbert) E.J. Gilbert contains the section *Amidella* (J. E. Gilbert) Konrad and Maubl., section *Arenariae* Zhu L. Yang, Y.Y. Cui and Q. Cai, section *Phalloideae* (Fr.) Quél., section *Roanokenses* Singer ex Singer, section *Strobiliformes* Singer ex Q. Cai, Zhu L. Yang and Y.Y. Cui, and section *Validae* (Fr.) Quél. The subgenus *Lepidella* Beauseigneur contains the section *Lepidella* Corner and Bas. Under this treatment, all saprotrophic *Amanita* species are assigned to the section *Lepidella*.

Surveys of species diversity, which can provide abundant materials, are fundamental and important work within mycological research. As more *Amanita* species are found and documented, researchers can have a better understanding and knowledge of this genus. However, the level of knowledge on the species diversity of Thai *Amanita* remains limited [25,26,27,28]. Through studying materials collected by their group over an extensive period and borrowed from other herbaria, Sanmee et al. [25] comprehensively reported 25 taxa of Thai *Amanita* with formal and detailed descriptions, including 18 records that were new to Thailand and seven known species. Between 2016 and 2018, Thongbai et al. [26,29,30] originally reported 12 species based on both morphology and phylogeny, which made an important contribution to the taxonomy of Thai *Amanita*. In addition, a number of researchers have worked on the species diversity of *Amanita* in Thailand [27,28,31,32,33,34,35], including those from our research group [33,34,35]. Up to now, 59 taxa have been reported from Thailand. Among these species, 21 were first described from Thailand and 24 belonged to the subgenus *Amanitina* according to the latest classification [25,26,28,29,30,31,32,33,34,35].

In the present study, we examined 20 specimens, which were collected from deciduous or coniferous forests predominantly composed of *Dipterocarpus*, *Shorea,* or *Pinus* species in northern and northeastern Thailand. On the basis of the macro- and microscopical characteristics, as well as multi-gene phylogenetic analyses, we identified them as nine *Amanita* species classified in the sections *Amidella*, *Phalloideae*, *Roanokenses*, and *Validae*. Among these taxa, three are reported as new to science, and four are, for the first time, reported from Thailand. 

## 2. Materials and Methods

### 2.1. Morphological Study

Basidiomata was collected from deciduous or coniferous forests in Chiang Mai, Chiang Rai, Phetchabun, and Sakon Nakho provinces in Thailand during the rainy season of 2019 and 2020. The following information was recorded: color photographs, forest type, substrate type, and geographic coordinates. Small pieces of tissue from the cap and/or stipe were taken and dried with silica gel to prepare for the molecular analyses [36], and the remaining specimens were dried at 35–45 °C for at least 12 h to prepare for the morphological examinations. All specimens observed in this study were deposited at the Herbarium of Biology Department (CMUB) and the Herbarium of Sustainable Development of Biological Resources (SDBR), Faculty of Science, Chiang Mai University, Thailand.

The macroscopic characteristics were described on the basis of field notes and images. The color codes and names were described according to Kornerup and Wanscher [37]. The microscopic features were observed in distilled water, 5% aqueous KOH (*w*/*v*), 1% Congo red (*w*/*v*), and Melzer’s reagent under a Leica DM500 microscope [18,19]. Sections of the pileipellis were cut along radial planes taken from halfway between the center and the margin of the pileus. Sections of the stipitipellis were cut longitudinally from small pieces taken from the middle part of the stipe. For the description of the basidiospores, the term (n/m/p) represents n basidiospores measured from m basidiomata of p collections. The dimensions for the basidiospores are given as (a–) b–c (–d), in which ‘b–c’ represents a minimum of 90% of the measured and extreme values, and ‘a’ and ‘d’ are given in parentheses whenever necessary. Q denotes the ratio of the length divided by the width of the basidiospore from the side view, Qm denotes the average Q of n measured basidiospores, and SD is their standard deviation. The results are presented as Q = Qm ± SD. Marginal striations on the pileus are expressed as a proportion of the ratio of the length of the striation to the radius of the pileus (nR). The terms denoting the basidiomata size and the spore shape are defined according to Bas [16] and Yang [19].

### 2.2. DNA Extraction, PCR Amplification, and Sequencing

Detailed processes of DNA extraction, PCR amplification, and sequencing protocols were carried out in line with previous studies [33,38]. Five DNA gene fragments were amplified and sequenced, including the internal transcribed spacer region (ITS), the large subunit of the nuclear ribosomal DNA (nrLSU), the partial sequences of the RNA polymerase II second largest subunit (*RPB2*), the translation elongation factor 1-alpha (*TEF1-α*), and the beta-tubulin gene (*TUB*). 

### 2.3. Phylogenetic Analyses

The newly generated sequences were used for BLAST searching in NCBI GenBank (https://www.ncbi.nlm.nih.gov, accessed on 7 March 2023), and then closely related sequences and Thai sequences were selected for the initial analysis. Detailed information, including the newly generated sequences and the sequences obtained from GenBank, is provided in Table 1. 

Sequences of each gene fragment were separately aligned with MAFFT v.7 [39] using the G-INSi iterative refinement algorithm and then manually optimized with AliView v.1.28 [40]. Gblocks v.0.91b [41] was used to exclude the ambiguously aligned regions for ITS with two options: “Allow smaller final blocks” and “Allow gap positions within the final blocks”. Sequence Matrix v.100.0 was applied to concatenate the five gene fragments for further phylogenetic analysis. MrModeltest v.2.3 [42] was adopted to determine the best fitting model of nucleotide substitution for each single-gene dataset by applying the default parameters. 

Phylogenetic trees were inferred using both maximum likelihood (ML) and Bayesian inference (BI), as detailed in [43]. The ML analysis was performed at the CIPRES web portal [44] using RAxML v.8.2.12 as part of the “RAxML-HPC BlackBox” tool [45] with the default settings, and the option “Estimate proportion of invariable sites (GTRGAMMA+I)” was set to “yes” for both the single-gene and the concatenated gene analyses. The phylogenetic analyses were initially performed on each single-gene alignment, and since there was no evident conflict (with ML bootstrap support of ≥75%), the concatenated dataset was built, and the multi-gene ML analysis was performed. The Bayesian analysis was carried out with MrBayes v.3.1.2 [46]. The posterior probabilities [47] were determined via Markov chain Monte Carlo sampling (MCMC) [48]. Six simultaneous Markov chains were run from random trees for one million generations, and the trees were sampled every 100th generation (the critical value for diagnosing topological convergence was 0.01). The first 25% of the trees were discarded, and the remaining trees were used for calculating the posterior probabilities in the majority-rule consensus tree. The phylogenetic trees were visualized with FigTree v.1.4.4 [49].

## 3. Results

### 3.1. Phylogenetic Analyses

The best fitting model for each gene fragment was as follows: general time reversible + proportion of invariable sites + gamma distribution (GTR + I + G) for nrLSU, *RPB2*, *TEF1-α*, and *TUB*; Hasegawa–Kishino–Yano (HKY) + I + G for ITS. The concatenated dataset was partitioned into five parts according to the sequence region. Because the model HKY + I + G could not be implemented in ML, the GTR + I + G model was used, as it included all the parameters of the selected model.

The multi-gene dataset comprised 466 sequences, of which 71 were newly generated and 395 were retrieved from GenBank. *Amanita flavofloccosa* (HKAS101443), *A. flavofloccosa* (HKAS90174), and *A. vittadinii* (HKAS101430) from *Amanita* section *Lepidella* were chosen as the outgroup taxa. The final multi-gene alignment comprised 3650 positions (nrLSU: 1–944; ITS: 945–2002; *RPB2*: 2003–2676; *TEF1-α*: 2677–3254; and *TUB*: 3255–3650), including gaps.

The resulting topologies of the ML and BI analyses were congruent; therefore, an ML tree is shown in Figure 1. In our phylogenetic analyses, all six sections of the subgenus *Amanitina* showed similar mutual relationships as those in previous studies [2,4], as well as the species in each section. The three novel species formed a clearly monophyletic lineage that was distinct from other extant species with credibly supported values.

### 3.2. Taxonomy

*Amanita* sect. *Amidella* (E. J. Gilbert) Konrad and Maubl., Agaricales: 61 (1948).

Basionym: *Amidella* E. J. Gilbert, in Bres., Iconogr. Mycol. 27 Suppl. 1(1): 71 (1940).

Type: *Amanita volvata* (Peck) Lloyd, Mycol. Writ. 1(7): 9 (1898).

Notes: Species from section *Amidella* have a series of remarkable characteristics, such as the color of the basidiomata changing to a brownish or reddish tone when injured, a striate and appendiculate pileal margin, the lamellae changing to a brown tone upon drying, truncate lamellulae, the amyloid basidiospores, and the absence of clamps [2,16,50]. The above combination of characteristics is unique to the section *Amidella* and is not found in any other section of *Amanita*. 

Presently, only two species of the section *Amidella* have been reported from Thailand, namely *Amanita avellaneosquamosa* (S. Imai) S. Imai and *A. clarisquamosa* (S. Imai) S. Imai [25]. In this study, six specimens collected from Chiang Mai and Phetchabun provinces were recognized and described as two novel species belonging to the section *Amidella*.

***Amanita claristriata*** Yuan S. Liu and S. Lumyong, sp. nov.; Figure 2a and Figure 3.

MycoBank number: 847954

Etymology: “*claristriata*”, from *clarus* (obvious) and *striatus* (grooved), indicates that this species has obvious striations on the margin of its pileus.

Holotype: THAILAND, Chiang Mai Province, Mueang District, 18°48′24.3″ N 98°54′38.1″ E, alt. 1102 m, 3 September 2020, Yuan S. Liu, STO-2020-404 (CMUB39992). GenBank accession numbers: OQ780686 (ITS), OQ780668 (nrLSU), OQ740048 (*RPB2*), and OQ740066 (*TEF1-α*).

*Basidiomata* medium-sized. *Pileus* 5.6–7.2 cm in diam., plano-convex to applanate, sometimes depressed at the center, white (1A1) to orange white (6A2); volval remnants on the pileus floccose to scaly, sometimes disappear because of rain, greyish orange (6B5–6) to brownish orange (7C5–6), densely arranged over the disk; margin inconspicuously striate at first and becoming obviously so with age, sometimes up to 0.3 R, appendiculate; context 5.0–7.0 mm wide, white (1A1), changing to orange white or pale orange (6A2–3) after injury. *Lamellae* free, crowded, white (1A1), becoming brown to dark brown (6F5–8) upon drying; lamellulae mostly truncate. *Stipe* 11.3–16.0 cm long × 0.7–1.3 cm diam. (the length includes the basal bulb), subcylindric or slightly tapering upwards, with the apex slightly expanded, white (1A1), covered with fibrous to floccose, white (1A1), greyish orange (6B3–4) to brown (6C4–6) squamules; context white (1A1), changing to orange white or pale orange (6A2–3) after injury, fistulose; basal bulb absent; volva saccate, 3.0–4.2 cm high × 1.8–2.4 cm wide, membranous, white (1A1) to greyish orange (6B5–6). *Annulus* absent. *Odor* not recorded. 

*Lamellar trama* bilateral. Mediostratum 30–40 μm wide, consisting of abundant clavate inflated cells (45–115 × 10–20 μm); filamentous hyphae abundant, 2–8 μm wide; vascular hyphae scarce. Lateral stratum 30–40 μm thick, consisting of abundant to dominated clavate inflated cells (30–72 × 9–16 μm), diverging at an angle of about 45° to the mediostratum; filamentous hyphae abundant, 2–6 μm wide. *Subhymenium* 20–30 μm thick, with two–three layers of subglobose or irregular cells, 9–23 × 6–16 μm. *Basidia* (Figure 3b) 33–47 × 8–11 μm, clavate, four-spored; sterigmata 3–6 μm long; basal septa lacking clamps. *Basidiospores* [82/3/3] (8.0–) 9.0–11.5 × 5.0–6.0 μm, avl × avw = 10.3 × 5.3 μm, Q = (1.60–) 1.64–2.20 (–2.30), Qm = 1.95 ± 0.18, elongate to cylindrical, thin-walled, smooth, colorless to pastel yellow, amyloid (Figure 3a). *Lamellar edge* sterile, consisting of subglobose to ellipsoid or clavate inflated cells (13–45 × 8–23 μm), single or in chains of 2–3, thin-walled, colorless to pastel yellow; filamentous hyphae abundant, 3–6 μm wide, irregularly arranged. *Pileipellis* 70–140 μm thick, two-layered; upper layer (30–80 μm thick) gelatinized, consisting of radially, thin-walled, colorless to pale yellow, filamentous hyphae 1–4 μm wide; lower layer (35–65 μm thick) consisting of radially and compactly arranged, filamentous hyphae 2–8 μm wide, colorless to pale yellow; vascular hyphae scarce. *Volval remnants* on pileus (Figure 3c) consisting of subradially to radially arranged elements: filamentous hyphae abundant, 2–10 μm wide, colorless pale yellow, thin-walled; inflated cells abundant, ellipsoid to clavate, 60–270 × 20–37 μm, colorless to pale yellow, thin-walled, often terminal; vascular hyphae scarce. Interior of *volval remnants* on stipe base consisting of sub-longitudinally to longitudinally arranged elements: filamentous hyphae abundant, 2–10 μm wide, colorless, thin-walled; inflated cells fairly abundant, subglobose to ellipsoid or ovoid, 40–98 × 15–55 μm, colorless, thin-walled, often terminal; vascular hyphae scarce. The outer surface of volval remnants on stipe base consisting of very abundant filamentous hyphae (2–6 μm wide), mixed with scattered to fairly abundant, subglobose to ellipsoid, or ovoid inflated cells. The inner surface of volval remnants on stipe base gelatinized, similar to structure of interior part but comprising much more filamentous hyphae (1–5 μm wide). *Stipe trama* consisting of longitudinally arranged, long clavate, terminal cells, 125–365 × 20–45 μm; filamentous hyphae scattered to abundant, 2–5 μm wide; vascular hyphae scarce. *Clamps* absent in all parts of basidioma.

Habitat: Solitary to scattered on soil in tropical deciduous forests dominated by *Dipterocarpus* and *Shorea* species. Basidiomata occurs in the rainy season.

Distribution: Currently known in northern Thailand.

Additional collections examined: THAILAND, Chiang Mai Province, Mueang District, alt. 1102 m, 3 September 2020, Yuan S. Liu, STO-2020-407 (SDBR-STO20-407); Yuan S. Liu, STO-2020-408 (SDBR-STO20-408).

Notes: *Amanita claristriata* is characterized by its medium-sized basidiomata, orange white pileus covered by floccose-felted to patchy, brownish orange volval remnants, an obvious striate margin on pileus (after maturity), the color of the basidiomata changes when injured (from white to pale orange), longer stipe (11.3–16.0 cm) covered by fibrous to floccose, brownish squamules, saccate volva remnants on the stipe base, as well as elongate to cylindrical basidiospores (9.0–11.5 × 5.0–6.0 μm, Qm = 1.95 ± 0.18). 

Morphologically, *Amanita lanigera* Y.Y. Cui, Q. Cai and Zhu L. Yang and *A. parvicurta* Y.Y. Cui, Q. Cai and Zhu L. Yang resemble *A. claristriata*. *Amanita lanigera*, described from China, differs from *A. claristriata* by having the non-striate pileal margin (or slightly striate), basidiomata color unchanged when injured, and larger, ellipsoid, colorless basidiospores (10.0–12.0 × 7.0–8.5 μm, Qm = 1.49 ± 0.13) [2].

Moreover, it is remarkable that *Amanita rufobrunnescens* W. Q. Deng and T. H. Li [51] and *A. volvata* [16,52,53] share a particular and consistent feature with *A. claristriata*, viz. the basidiomata changes to light red or pale orange after injury. However, both *A. rufobrunnescens* reported from China and *A. volvata* reported from America have larger basidiospores (10.0–12.0 × 5.5–6.5 μm, Qm = 1.78 ± 0.17 for *A. rufobrunnescens*; 10.0–12.5 × 6.0–7.5 μm, Qm = 1.67 ± 0.11 for *A. volvata*) [2,16,51,53].

Phylogenetically, *Amanita claristriata* is related to *A. peckiana* Kauffman and *A. pinophila* Y.Y. Cui, Q. Cai and Zhu L. Yang. However, the latter two species differ from the former by not changing basidiomata color when injured, as well as larger basidiospores (9.8–13.6 × 5.6–7.0 μm for *A. peckiana*; 10.0–12.0 × 5.5–7.0 μm, Qm = 1.81 ± 0.14 for *A. pinophila*) [2,53]. 

***Amanita fulvisquamea*** Yuan S. Liu and S. Lumyong, sp. nov.; Figure 2b–d and Figure 4. 

MycoBank number: 847955

Etymology: “*fulvisquamea*”, from *fulvus* (brownish) and *squameus* (covered with scales), referring to the brown scales on its pileus.

Holotype: THAILAND, Phetchabun Province, Nam Nao District, 16°42′37″ N 101°35′55″ E, alt. 870 m, 21 August 2020, Yuan S. Liu, STO-2020-367 (CMUB39993). GenBank accession numbers: OQ780689 (ITS), OQ780671 (nrLSU), OQ740051 (*RPB2*), OQ740069 (*TEF1-α*), and OQ740087 (*TUB*).

*Basidiomata* small- to medium-sized. *Pileus* 3.0–5.8 cm in diam., plano-convex to applanate, white (1A1) with pale orange (5A3) tone; volval remnants on pileus floccose to scaly, white (1A1), greyish orange (6B4–6) to brown (6C4–6), densely arranged over the disk; margin inconspicuously striate at first and becoming obviously so with age, appendiculate; context 3.5–7.0 mm wide, white (1A1), unchanging. *Lamellae* free, crowded, white (1A1), becoming greyish orange (5B3–5) to brown (6E5–8) upon drying; lamellulae mostly truncate. *Stipe* 6.0–7.5 cm long × 0.5–1.2 cm diam. (the length includes the basal bulb), subcylindric or slightly tapering upwards, with the apex slightly expanded, white (1A1), covered with floccose, white (1A1), greyish orange (6B4–6) to brown (6C4–6) squamules; context white (1A1), unchanging, fistulose; basal bulb absent; volva saccate, 2.5–3.1 cm high × 1.8–2.7 cm wide., membranous, white (1A1) to brown (6C4–6). *Annulus* present, white (1A1), fugacious. *Odor* not recorded.

*Lamellar trama* bilateral. Mediostratum 25–40 μm wide, consisting of abundant clavate to oblong inflated cells (35–145 × 12–35 μm); filamentous hyphae abundant, 2–7 μm wide; vascular hyphae scarce. Lateral stratum 35–55 μm thick, consisting of abundant oblong inflated cells (22–75 ×11–32 μm), diverging at an angle of about 45° to the mediostratum; filamentous hyphae abundant, 3–8 μm wide. *Subhymenium* 25–35 μm thick, with two–three layers of subglobose or irregular cells, 9–25 × 7–13 μm. *Basidia* (Figure 4b) 33–51 × 10–14 μm, clavate, four-spored; sterigmata 3–5 μm long; basal septa lacking clamps. *Basidiospores* [81/3/3] (8.0–) 8.5–11.0 (–11.5) × (6.5–) 7.0–8.0 (–8.5) μm, avl × avw = 9.7 × 7.2 μm, Q = (1.13–) 1.20–1.57 (–1.64), Qm = 1.35 ± 0.11, broadly ellipsoid to ellipsoid, sometimes subglobose or elongate, thin-walled, smooth, colorless to pale yellow or dull yellow, amyloid (Figure 4a). *Lamellar edge* sterile, consisting of subglobose to ellipsoid or pyriform inflated cells (18–65 × 11–30 μm), single or in chains of two–three, thin-walled, colorless; filamentous hyphae abundant, 2–5 μm wide, irregularly arranged. *Pileipellis* 110–170 μm thick, two-layered; upper layer (80–135 μm thick) strongly gelatinized, consisting of subradially arranged, thin-walled, colorless to pale yellow, filamentous hyphae 1–8 μm wide; lower layer (20–35 μm thick) consisting of radially and compactly arranged, filamentous hyphae 2–6 μm wide, colorless to pale yellow; vascular hyphae scarce. *Volval remnants* on pileus (Figure 4c) consisting of subradially to radially arranged elements: filamentous hyphae abundant, 2–8 μm wide, colorless to pale yellow, thin-walled; inflated cells abundant, ellipsoid to clavate, 43–200 × 13–37 μm, colorless to pale yellow, thin-walled, often terminal; vascular hyphae scarce. Interior of *volval remnants* on stipe base consisting of sub-longitudinally arranged elements: filamentous hyphae abundant, 2–12 μm wide, colorless, thin-walled; inflated cells abundant, subglobose to ellipsoid, 30–50 × 12–40 μm, colorless, thin-walled, often terminal; vascular hyphae scarce. The outer surface of volval remnants on stipe base consisting of very abundant filamentous hyphae (1–11 μm wide), mixed with scattered to fairly abundant, subglobose to ellipsoid, or ovoid to pyriform inflated cells. The inner surface of volval remnants on stipe base gelatinized, similar to structure of interior part but comprising much more filamentous hyphae (1–10 μm wide). *Stipe trama* consists of longitudinally arranged, long clavate terminal cells, 105–320 × 16–35 μm; filamentous hyphae scattered to abundant, 2–10 μm wide; vascular hyphae scarce. *Clamps* absent in all parts of basidioma.

Habitat: Solitary to scattered on soil in tropical deciduous or coniferous forests dominated by *Dipterocarpus*, *Shorea*, and *Pinus* species. Basidiomata occurs in the rainy season.

Distribution: Currently known in northern Thailand.

Additional collections examined: THAILAND, Chiang Mai Province, Mae On District, alt. 1201 m, 30 July 2020, Yuan S. Liu, STO-2020-211 (SDBR-STO20-211). Phetchabun Province, Nam Nao District, alt. 870 m, 21 August 2020, Yuan S. Liu, STO-2020-377 (SDBR-STO20-377).

Notes: *Amanita fulvisquamea* is characterized by its small- to medium-sized basidiomata, pulverulent to floccose, or patchy and greyish orange to brown volval remnants on pileus, striate pileal margin, saccate volva remnants on the stipe base, as well as broadly ellipsoid to ellipsoid basidiospores (8.5–11.0 × 7.0–8.0 μm, Qm = 1.35 ± 0.11). 

Morphologically, *Amanita fulvisquamea* is easily confused with *A. clarisquamosa*, *A. parvicurta*, and *A. volvata* due to the similar appearances, e.g., floccose to pulverulent, brownish volval remnants on the pileus; floccose, brownish squamules on the stipe, as well as saccate volva remnants on the stipe base. However, *A. clarisquamosa* has much longer basidiospores, which mainly possess elongate shape, while both *A. parvicurta* and *A. volvata* have much narrower and elongate-shaped basidiospores [2,16,53]. 

Phylogenetically, *Amanita brunneomaculata* Y.Y. Cui, Q. Cai and Zhu L. Yang is closely related to *A. fulvisquamea*. However, the former differs from the latter by having a distinctly spotted pileus, as well as much longer basidiospores (10.0–13.0 × 6.5–8.0 μm, Qm = 1.65 ± 0.19) [2]. 

*Amanita lanigera* is morphologically similar and phylogenetically related to *A. fulvisquamea*. However, *A. lanigera*, originally reported from China, has thicker pileipellis (75–230 μm), larger inflated terminal cells (80–520 × 15–45 μm) in the stipe trama, as well as longer, ellipsoid, and colorless basidiospores (10.0–12.0 × 7.0–8.5 μm, Qm = 1.35 ± 0.11) [2].

***Amanita* sect. *Phalloideae*** (Fr.) Quél., Mém. Soc. Emul. Montbéliard, Ser. II, 5: 66 (1872).

Basionym: *Amanita* [sect.] *Phalloideae* Fr., Monogr. Amanit. Sueciae: 3 (1854).

Lectotype: *Amanita phalloides* (Vaill. ex Fr.) Link., Handbuch zur Erkennung der Nutzbarsten und am häufigsten vorkommenden Gewächse: 272 (1833).

Notes: According to previous studies, *Amanita* sect. *Phalloideae* phylogenetically comprises three subclades that are well supported in phylogenetic analyses and by morphological evidence. Our multi-locus phylogenetic analysis also presented the same result. These three subclades may be treated as subsections or new sections [2,4,26,54] in the future.

Up to now, six taxa of section *Phalloideae* have been reported from Thailand, namely *Amanita ballerina* Raspé, Thongbai and K.D. Hyde, *A. brunneitoxicaria* Thongbai, Raspé and K.D. Hyde, *A. fuliginea* Hongo, *A. fuligineoides* P. Zhang and Zhu L. Yang, *A. rimosa* P. Zhang and Zhu L. Yang and *A. zangii* Zhu L. Yang, T.H. Li and X.L. Wu [25,26,29]. One more taxon is recognized in our phylogenetic analysis, and here it is described as a new species based on morphological evidence as well.

***Amanita albifragilis*** Yuan S. Liu and S. Lumyong, sp. nov.; Figure 2e–f and Figure 5.

MycoBank number: 847956

Etymology: “*albifragilis*”, from *albus* (whitish) and *fragilis* (brittle), refers to the white fruiting body and the thin and brittle surface of the pileus.

Holotype: THAILAND, Sakon Nakhon Province, Kut Bak District, 17°6′4″ N 103°54′32″ E, alt. 205 m, 15 August 2020, Yuan S. Liu, STO-2020-300 (CMUB39994). GenBank accession numbers: OQ780692 (ITS), OQ780674 (nrLSU), and OQ740072 (*TEF1-α*).

*Basidiomata* small- to medium-sized. *Pileus* 3.2–5.2 cm in diam., plano-convex to applanate, often depressed at center, surface thin and fragile, white (1A1); volval remnants on pileus absent; margin non-striate, non-appendiculate; context 1.5–2.5 mm wide, white (1A1), unchanging. *Lamellae* free, crowded, white (1A1); lamellulae mostly truncate. *Stipe* 5.4–8.0 cm long × 0.5–0.6 cm diam. (the length includes the basal bulb), subcylindric or slightly tapering upwards, with apex slightly expanded, white (1A1), covered with fibrous, white (1A1), squamules; context white (1A1), unchanging, fistulose to solid; basal bulb subglobose; volva limbate, 1.3–1.8 cm high × 1.4–1.8 cm wide., membranous, white (1A1). *Annulus* subapical, membranous, white (1A1), persistent. *Odor* not recorded.

*Lamellar trama* bilateral. Mediostratum 20–30 μm wide, consists of abundant ellipsoid to elongate inflated cells (60–108 × 15–32 μm); filamentous hyphae abundant, 2–8 μm wide; vascular hyphae scarce. The lateral stratum consists of abundant ellipsoid to clavate inflated cells (30–60 × 11–22 μm), diverging at an angle of about 45° to the mediostratum; filamentous hyphae abundant, 2–6 μm wide. *Subhymenium* 20–30 μm thick, with two–three layers of subglobose to ellipsoid or irregular cells, 7–19 × 6–13 μm. *Basidia* (Figure 5b) 28–45 × 8–12 μm, clavate, four-spored; sterigmata 3–6 μm long; basal septa lacking clamps. *Basidiospores* [69/2/2] (7.0–) 8.0–9.0 (–10.5) × 6.0–7.0 (–9.0) μm, avl × avw = 8.5 × 6.8 μm, Q = (1.13–) 1.14–1.39 (–1.50), Qm = 1.26 ± 0.09, broadly ellipsoid, sometimes subglobose, thin-walled, smooth, colorless, and amyloid (Figure 5a). *Lamellar edge* sterile, consisting of subglobose to ellipsoid or clavate inflated cells (14–50 × 12–28 μm), single or in chains of two–three, thin-walled, colorless; filamentous hyphae scattered, 1–3 μm wide, irregularly arranged. *Pileipellis* 60–110 μm thick, two-layered; upper layer (20–40 μm thick) slightly gelatinized, consisting of subradially arranged, thin-walled, colorless, filamentous hyphae 2–8 μm wide; lower layer (30–70 μm thick) consisting of radially and compactly arranged, filamentous hyphae 2–8 μm wide, colorless; vascular hyphae scarce. The interior of *volval remnants* on the stipe base consists of sub-longitudinally to irregularly arranged elements: filamentous hyphae very abundant, 3–11 μm wide, colorless, thin-walled; inflated cells scarce to scattered, clavate, 70–110 × 10–36 μm, colorless, thin-walled; vascular hyphae scarce. The outer surface of volval remnants on stipe base (Figure 5c) predominately consists of very abundant filamentous hyphae (3–12 μm wide), mixed with scarce, clavate inflated cells. The inner surface of volval remnants on the stipe base gelatinized, similar to structure of interior part but comprising much more filamentous hyphae (3–9 μm wide). *Stipe trama* consists of longitudinally arranged, abundant, long clavate terminal cells (80–285 × 12–23 μm); filamentous hyphae abundant to very abundant, 2–9 μm wide; vascular hyphae scarce. *Annulus* consists of radially arranged elements: inflated cells scattered, clavate, often terminal, 35–75 × 9–17 μm, colorless, thin-walled; filamentous hyphae very abundant to predominant, 2–9 μm wide, colorless, thin-walled; vascular hyphae scarce. *Clamps* absent in all parts of basidioma.

Habitat: Solitary to scattered on soil in tropical deciduous forests dominated by *Dipterocarpus* and *Shorea* species. Basidiomata occurs in the rainy season.

Distribution: Currently known in northeastern Thailand.

Additional collections examined: THAILAND, Sakon Nakhon Province, Kut Bak District, alt. 205 m, 15 August 2020, Yuan S. Liu, STO-2020-304 (SDBR-STO20-304).

Notes: *Amanita albifragilis* is characterized by its small- to medium-sized basidiomata, slightly depressed pileal center, the thin and fragile surface of the pileus, a non-striate pileal margin, subglobose stipe base surrounded by limbate volva remnants, subapical and persistent annulus, as well as the broadly ellipsoid amyloid basidiospores (8.0–9.0 × 6.0–7.0 μm, Qm = 1.26 ± 0.09).

At first sight, *Amanita rimosa*, originally reported from China [55], resembles the newly described species rather strongly. Both species share a number of similar or identical features, e.g., a small and white basidioma, a smooth pileus with slightly rimose margin, the limbate volva remnants on the stipe base, and a membranous and persistent annulus [2,19,55]. However, *A. rimosa* has globose to subglobose basidiospores (7.0–8.5 × 6.5–8.0 μm, Qm = 1.08 ± 0.05) and a different structure of pileipellis containing more abundant ellipsoid to clavate inflated cells [2,19,55].

Except for *Amanita rimosa*, there are a number of taxa that have white basidiomata in section *Phalloideae*, e.g., *A. exitialis* Zhu L. Yang and T. H. Li, *A. parviexitialis* Q. Cai, Zhu L. Yang and Y.Y. Cui, *A. virosa* Bertillon and *A. subjunquillea* S. Imai. Among the above taxa, *A. parviexitialis* is easily confused with *A*. *albifragilis* due to its small basidioma, a smooth and depressed pileus, and the limbate volva remnants on the stipe base. However, *A. parviexitialis* usually has brownish tone in the pileal center, two-spored basidia, as well as much wider, subglobose basidiospores (7.5–9.5 × 7.0–9.0 μm, Qm = 1.09 ± 0.05) [2,19,56].

Phylogenetically, *Amanita albifragilis* is closely related to *A. griseorosea* Q. Cai, Zhu L. Yang and Y.Y. Cui and *A. molliuscula* Q. Cai, Zhu L. Yang and Y.Y. Cui. *Amanita griseorosea* can be easily distinguished from *A. albifragilis* by having a grayish-brown pileus with dark-gray fibrils and pinkish lamellae [2,19,38,56]. Compared to the newly described species, *A. molliuscula* has much more abundantly inflated cells in structures of pileipellis and in the interior of volval remnants on stipe base, as well as the wider and globose to subglobose basidiospores (7.5–9.0 × 7.0–8.0 μm, Qm = 1.07 ± 0.06) [2,38,56].

***Amanita* sect. *Roanokenses*** Singer ex Singer, Sydowia 15: 67 (1962).

Synonym: *Amanita* subsect. *Limbatulae* Bas, Persoonia 5: 528 (1969).

Type: *Amanita roanokensis* Coker, J. Elisha Mitchell scient. Soc. 43: 141 (1927).

Notes: *Amanita* sect. *Roanokenses* is one of the most species-diverse sections in *Amanita* subgen. *Amanitina*. To date, nine species have been reported from Thailand, namely *A. alboflavescens* Hongo, *A. atrobrunnea* Thongbai, Raspé and K.D. Hyde, *Amanita* cf. *oberwinkleriana*, *A. hongoi* Bas, *A. japonica* Hongo ex Bas, *A. macrocarpa* W. Q. Deng, T. H. Li and Zhu L. Yang, *A. manginiana* sensu W.F. Chiu, *A. pseudoporphyria* Hongo and *A. virgineoides* Bas [2,25,26,28]. In this study, four taxa belonging to the section *Roanokenses* were recognized and are presented below. 

***Amanita caojizong*** Zhu L. Yang, Y.Y. Cui and Q. Cai, Fungal Divers. 91: 138 (2018). Figure 2g and Figure 6.

*Basidiomata* large. *Pileus* 9.5–12.0 cm diam., convex to plano-convex, milk white to greyish yellow (1B2–4) or greyish brown (5E3), possessing innate dark-grey radiating fibrils; volval remnants on pileus often absent; margin non-striate, appendiculate; context 8–9 mm wide, white (1A1), unchanging. *Lamellae* free, crowded, white (1A1); lamellulae attenuate. *Stipe* 14.3–20.0 cm long × 1.5–1.7 cm diam. The length includes the basal bulb, cylindrical or slightly tapering upwards with apex slightly expanded, white (1A1), covered with fibrous squamules; context solid, white (1A1); basal part 2.2–3.2 cm diam., fusiform to clavate; volval remnants on stipe base sheathed, membranous, with free limb up to 6.1 cm high, white (1A1). *Annulus* apical, white, fragile, and fugacious when mature. *Odor* not recorded.

*Lamellar trama* bilateral. Mediostratum 20–40 μm wide, consisting of abundant ellipsoid to elongate inflated cells (53–90 × 13–22 μm); filamentous hyphae abundant, 2–6 μm wide; vascular hyphae scarce. Lateral stratum 20–30 μm wide, consisting of abundant elongate to clavate inflated cells (36–65 × 12–22 μm), diverging at an angle of about 45° to the mediostratum; filamentous hyphae abundant, 3–5 μm wide. *Subhymenium* 20–30 μm thick, with two–three layers of subglobose, ovoid to ellipsoid, or irregular cells, 6–25 × 5–16 μm. *Basidia* (Figure 6b) 32–45 × 8–10 μm, clavate, four-spored; sterigmata up to 4–5 μm long; basal septa lacking clamps. *Basidiospores* [75/3/3] (6.0–) 6.5–8.0 (–9.0) × 5.0–7.0 μm, avl × avw = 7.5 × 6.0 μm, Q = (1.00–) 1.14–1.36 (–1.50) μm, Qm = 1.25 ± 0.10, broadly ellipsoid, sometimes globose to subglobose, thin-walled, smooth, colorless, amyloid (Figure 6a). *Lamellar edge* sterile, consisting of subglobose to ellipsoid inflated cells (9–17 × 8–14 μm), single or in chains of two–three, thin-walled, colorless; filamentous hyphae abundant, 2–7 μm wide, irregularly arranged. *Pileipellis* 90–150 μm thick, two-layered; upper layer (30–90 μm thick) strongly gelatinized, consisting of radially, thin-walled, colorless or light brownish, filamentous hyphae 2–5 μm wide; lower layer (45–110 μm thick) consisting of radially and compactly arranged filamentous hyphae 2–7 (–12) μm wide, yellowish to brownish; vascular hyphae scarce. The inner part of *volval remnants* on the stipe base consists of longitudinally arranged elements: filamentous hyphae predominant, 2–8 μm wide, colorless, thin-walled, branching; inflated cells scarce to scattered, ellipsoid to clavate, sometimes subglobose, 65–125 × 13–30 μm, colorless, thin-walled, interjacent, or terminal; vascular hyphae scarce. Th outer surface of volval remnants on the stipe base (Figure 6c) is similar to the inner part but with more abundant filamentous hyphae. The inner surface of volval remnants on the stipe base is similar to the interior part but slightly gelatinized. *Stipe trama* consists of longitudinally arranged, long clavate terminal cells, 130–220 × 15–30 μm; filamentous hyphae abundant, 2–10 μm wide; vascular hyphae scarce. *Annulus* consists of loosely arranged, interwoven elements: inflated cells abundant, globose, subglobose to pyriform, 15–55 × 13–48 μm, colorless, thin-walled; filamentous hyphae fairly abundant, 1–6 μm wide, colorless, thin-walled; vascular hyphae scarce. *Clamps* absent in all parts of basidioma.

Habitat: Solitary to scattered on soil in tropical deciduous forests dominated by *Dipterocarpus* and *Shorea* species. Basidiomata occurs in the rainy season.

Distribution: This species is currently known in China [2,57], Japan [58], Korea [59], and Thailand ([26], this study).

Specimens examined: THAILAND, Chiang Mai Province, Mae Taeng District, alt. 720 m, 9 August 2019, Yuan S. Liu, STO-2019-473 (SDBR-STO19-473); Mae On District, alt. 753 m, 6 July 2020, Yuan S. Liu, STO-2020-120 (SDBR-STO20-120). Chiang Rai Province, Mae Fa Luang District, alt. 1236 m, 10 July 2020, Yuan S. Liu, STO-2020-169 (SDBR-STO20-169). 

Notes: *Amanita caojizong*, reported from China, is a common edible mushroom found in Yunnan province. It is morphologically similar to a number of taxa, such as *A. pseudoporphyria*, *A. pseudomanginiana* Q. Cai, Y.Y. Cui and Zhu L. Yang, *A. griseoturcosa* T. Oda, C. Tanaka and Tsuda, *A. roseolifolia* Y.Y. Cui, Q. Cai and Zhu L. Yang and *A. modesta* Corner and Bas. Detailed comparisons between *A. caojizong* and these similar species can be found in Cui et al. [2]. It is worth noting that our Thai collections had a much wider color range on the pileus, i.e., milk white to greyish yellow or greyish brown.

***Amanita griseofarinosa*** Hongo, Mem. Fac. Lib. Arts Shiga Univ. Nat. Sci. 11: 39 (1961). Figure 2h and Figure 7.

*Basidiomata* small- to medium-sized. *Pileus* 3.5–6.5 cm diam., convex to plano-convex, or applanate to plano-concave, light grey (4C1–2) to brownish grey (4D1–2); volval remnants on pileus floccose to pulverulent, brownish grey (4D2–3), yellowish (5D3–5) to yellowish brown (5E4–5), densely arranged on the disc; margin non-striate, appendiculate; context 3–4.5 mm wide, white (1A1), unchanging. *Lamellae* free, crowded, white (1A1); lamellulae attenuate. *Stipe* 7.2–12.0 cm long × 0.5–1.2 cm diam. (the length includes the basal bulb), cylindrical, densely covered by floccose to pulverulent yellowish-white (4A2) to orange-white (5A2) squamules; context stuffed, white (1A1); basal part 0.9–1.8 cm diam., clavate to ventricose, upper part covered with floccose to pulverulent, yellowish-white (4A2) to orange-white (5A2) volval remnants. *Annulus* fragile and fugacious. *Odor* not recorded.

*Lamellar trama* bilateral. Mediostratum 25–35 μm wide, consisting of abundant ellipsoid inflated cells (45–95 × 10–22 μm); filamentous hyphae abundant, 2–7 (–11) μm wide; vascular hyphae scarce. Lateral stratum 20–30 μm wide, consisting of abundant clavate inflated cells (35–85 × 8–25 μm), diverging at an angle of about 45° to the mediostratum; filamentous hyphae abundant, 3–7 μm wide. *Subhymenium* 20–30 μm thick, with two–three layers of subglobose to ellipsoid or irregular cells, 6–20 × 6–12 μm. *Basidia* (Figure 7b) 40–56 × 11–13 μm, clavate, four-spored; sterigmata up to 3–5 μm long; basal septa lacking clamps. *Basidiospores* [68/2/2] (7.0–) 7.5–10.0 (–11.0) × (6.0–) 6.5–8.5 (–9.0) μm, avl × avw = 8.6 × 7.4 μm, Q = (1.00–)1.06–1.31 (–1.33) μm, Qm = 1.17 ± 0.09, mainly subglobose to broadly ellipsoid, sometimes globose or ellipsoid, thin-walled, smooth, colorless, amyloid (Figure 7a). *Lamellar edge* sterile and consists of subglobose to ellipsoid inflated cells (15–33 × 10–30 μm), single or in chains of two–three, thin-walled, colorless; filamentous hyphae abundant, 2–6 μm wide, irregularly arranged. *Pileipellis* 50–100 μm thick, consisting of radially and compactly, thin-walled, colorless to light brownish, filamentous hyphae 2–5 μm wide, with its outer-surface hyphae loosely and irregularly arranged; vascular hyphae scarce to scattered. *Volval remnants* on pileus (Figure 7c) composed of abundant irregularly arranged filamentous hyphae 2–7 μm wide, mixed with abundant to predominant subglobose, or broadly clavate to fusiform inflated cells (15–53 × 10–40 μm). *Volval remnants* on stipe base similar to structure of volval remnants on pileus, predominately composed of irregularly arranged subglobose, or broadly clavate to fusiform inflated cells (9–50 × 8–32 μm), mixed with abundant filamentous hyphae (2–7 μm wide). *Stipe trama* consists of abundant longitudinally arranged, long clavate terminal cells, 80–220 × 19–30 μm; filamentous hyphae abundant, 3–10 μm wide; vascular hyphae scarce. *Clamps* absent in all parts of basidioma.

Habitat: Solitary to scattered on soil in tropical deciduous forests dominated by *Dipterocarpus* and *Shorea* species. Basidiomata occurs in the rainy season.

Distribution: This species is currently known in China [2,18,19], Japan [16,60,61], Korea [62], and Thailand [this study].

Specimens examined: THAILAND, Chiang Mai Province, Mae Taeng District, alt. 1077 m, 31 May 2020, Yuan S. Liu, STO-2020-8 (SDBR-STO20-08); Yuan S. Liu, STO-2020-9 (SDBR-STO20-09).

Notes: *Amanita griseofarinosa* was first reported from Japan [60] and was then found in China and Korea [2,18,19,62]. Our two specimens possess small- to medium-sized basidiomata, pileus and stipe densely covered by floccose to pulverulent, brownish-grey or yellowish-white squamules, appendiculate margin, attenuate lamellulae, and fragile and fugacious annulus. All these features are consistent with the type specimen. 

Morphologically, the species is similar to *Amanita* cf. *griseofarinosa* (HKAS 79587) and *A. vestita* Corner and Bas on account of brownish-gray basidiomata, and pulverulent volval remnants densely pervade the surface of the pileus and stipe [2,15,19]. However, *Amanita* cf. *griseofarinosa* differs from *A. griseofarinosa* in its original sense by having clamps, as well as the much wider and globose to subglobose basidiospores (8.5–10.5 × 8.5–10.0 μm, Qm = 1.04 ± 0.04) [2,19]. Moreover, *Amanita* cf. *griseofarinosa* is phylogenetically distinct from *A. griseofarinosa* [2]. *Amanita vestita*, reported from Singapore, is distinguished from *A*. *griseofarinosa* by having small basidiomata, slightly depressed pileal center, and much narrower basidiospores (7.5–9.0 × 5.5–6.5 μm) [15].

Undoubtedly, *Amanita berkeleyi* (Hook. f.) Bas, originally described from India, is closely related to *A. griseofarinosa* [16]. However, *A. berkeleyi* possesses large to very large basidiomata, felted-pulverulent to crust-like volva remnants on its pileus, and much wider basidiospores (8.0–10.5 × 6.5–9.5 μm) [16].

*Amanita cinereovelata* Hosen, reported from Bangladesh [63], is phylogenetically related and morphologically similar to *A. griseofarinosa*. However, *A. cinereovelata* differs by having a thicker pileipellis (up to 290 μm), globose to subglobose basidiospores (9.0–10.0 × 8.0–9.0 μm, Qm = 1.12 ± 0.05), and the presence of clamps [63].

***Amanita neoovoidea*** Hongo, Mem. Shiga Univ. 25: 57 (1975). Figure 2i and Figure 8.

*Basidiomata* large. *Pileus* 9.0–10.0 cm diam., convex to plano-convex, white (1A1) to yellowish white (1A2); volval remnants on pileus consisting of two layers: outer layer membranous, yellowish white to pale yellow (1A2–3); inner layer floccose to pulverulent, white (1A1); margin non-striate, appendiculate; context 8.5–10.0 mm wide, white (1A1), unchanging. *Lamellae* free, crowded, white (1A1); lamellulae attenuate. *Stipe* 12.0–13.5 cm long × 1.5–1.6 cm diam. (the length includes the basal bulb), cylindrical, densely covered by floccose to pulverulent white (1A1) squamules; context solid, white (1A1); basal part 2.6–3.0 cm diam., fusiform to ventricose; volval remnants on stipe base yellowish white (3A2), arranged in incomplete belts or with a recurved friable limb. *Annulus* subapical, white (1A1), fragile and fugacious. *Odor* not recorded.

*Lamellar trama* bilateral. Mediostratum 25–40 μm wide, consisting of abundant clavate to fusiform inflated cells (35–110 × 12–20 μm); filamentous hyphae abundant, 2–6 μm wide; vascular hyphae scarce. Lateral stratum 20–30 μm wide, consisting of abundant elongate to clavate inflated cells (20–145 × 12–30 μm), diverging at an angle of about 45° to the mediostratum; filamentous hyphae abundant, 3–7 μm wide. *Subhymenium* 20–30 μm thick, with two–three layers of subglobose, ovoid to ellipsoid or irregular cells, 6–27 × 6–16 μm. *Basidia* (Figure 8b) 40–55 × 10–12 μm, clavate, four-spored; sterigmata up to 3–5 μm long; basal septa lacking clamps. *Basidiospores* [52/2/2] 7.0–10.0 × 5.0–7.0 μm, avl × avw = 8.5 × 6.1 μm, Q = (1.14–)1.21–1.67 (–1.70) μm, Qm = 1.41 ± 0.12, broadly ellipsoid, sometimes subglobose or elongate, thin-walled, smooth, colorless, amyloid (Figure 8a). *Lamellar edge* sterile, consisting of pyriform to subglobose or clavate inflated cells (27–50 × 18–36 μm), single or in chains of two–three, thin-walled, colorless; filamentous hyphae abundant, 2–5 μm wide, irregularly arranged. *Pileipellis* 45–120 μm thick, consisting of radial, strongly gelatinized, colorless filamentous hyphae (2–6 μm wide); vascular hyphae scarce. *Volval remnants* on pileus (Figure 8c) consist of two layers. The outer layer of volval remnants on the pileus consists of more or less radially arranged elements: inflated cells (45–173 × 15–32 μm) fairly abundant to abundant; filamentous hyphae (2–9 μm wide) very abundant; vascular hyphae scarce. The inner layer of volval remnants on the pileus consists of irregularly arranged elements: inflated cells (14–83 × 10–50 μm) abundant to predominant, subglobose to clavate; filamentous hyphae (3–8 μm wide) abundant; vascular hyphae scarce. *Volval remnants* on stipe base composed of irregularly to vertical-arranged elements: inflated cells (36–105 × 16–25 μm) fairly abundant to abundant; filamentous hyphae (1.5–7.0 μm wide) very abundant to dominate; vascular hyphae scarce. *Stipe trama* consists of longitudinally arranged, long clavate terminal cells, 180–240 × 13–25 μm; filamentous hyphae abundant, 3–10 μm wide; vascular hyphae scarce. *Annulus* consists of loosely, irregularly arranged elements: inflated cells abundant, subglobose, ellipsoid to clavate, 23–70 × 15–32 μm, colorless, thin-walled; filamentsous hyphae scarce to fairly abundant, 1.5–5.0 μm wide, colorless, thin-walled; vascular hyphae scarce. *Clamps* absent in all parts of basidioma.

Habitat: Solitary to scattered on soil in tropical deciduous forests dominated by *Dipterocarpus* and *Shorea* species. Basidiomata occurs in the rainy season.

Distribution: This species is currently known in China [2,17,18,19], Nepal [64], Japan [64,65], Korea [66], and Thailand (this study).

Specimens examined: THAILAND, Chiang Mai Province, Mae On District, alt. 704 m, 6 July 2020, Yuan S. Liu, STO-2020-110 (SDBR-STO20-110); Mae Taeng District, alt. 720 m, 9 August 2019, Yuan S. Liu, STO-2019-503 (SDBR-STO19-503).

Notes: *Amanita neoovoidea*, originally described from Japan [65], is characterized by medium-sized to large basidiomata, pileal volval remnants arranged in two layers, with the outer layer being membranous and inner layer being floccose to pulverulent, appendiculate pileal margin, stipe densely covered by floccose to pulverulent squamules, with incomplete belts or recurved friable limb remnants on the stipe base, and a fragile and fugacious annulus. Our Thai materials are consistent with all the features above.

Morphologically, *Amanita duplex* Corner and Bas, reported from Singapore, is undoubtedly similar to this species. Moreover, in our phylogenetic analysis, *A. neoovoidea* is related to *A. pseudomanginiana*, *A. pseudoporphyria*, and *A. atrobrunnea*. Detailed comparisons between *A. neoovoidea* and the four related species above can be found in previous studies [2,18].

***Amanita oberwinkleriana*** Zhu L. Yang and Yoshim. Doi, Bull. Natn. Sci. Mus. Tokyo 25 (3): 120 (1999). Figure 2j and Figure 9.

*Basidiomata* small- to medium-sized. *Pileus* 4.5–6.0 cm diam., plano-convex to applanate, sometimes plano-concave, smooth, white (1A1), often yellowish white (1A2) to pale yellow (4A2–3) in the center; volval remnants on pileus often absent; margin non-striate, sometimes with inconspicuous stripes (ca. 0.2–0.3 R), non-appendiculate; context 2.0–4.0 mm wide, white (1A1), unchanging. *Lamellae* free, crowded, white to yellowish white (1A1–2); lamellulae attenuate. *Stipe* 8.2–9.5 cm long × 0.7–1.0 cm diam. (the length includes the basal bulb), cylindrical, covered by white (1A1), fibrous to tomentose squamules; context fistulose, white (1A1); basal part 1.6–2.0 cm diam., fusiform to napiform; volval remnants on stipe base limbate, membranous, with free limb up to ca. 1.5 cm high, both surfaces white (1A1). *Annulus* subapical, membranous, white (1A1). *Odor* not recorded.

*Lamellar trama* bilateral. Mediostratum 25–30 μm wide, consisting of abundant ellipsoid inflated cells (48–105 × 13–23 μm); filamentous hyphae abundant, 3–8 μm wide; vascular hyphae scarce. Lateral stratum 25–30 μm wide, consisting of abundant ellipsoid to clavate inflated cells (32–60 × 12–25 μm), diverging at an angle of about 45° to the mediostratum; filamentous hyphae abundant, 3–7 μm wide. *Subhymenium* 30–40 μm thick, with two–three layers of subglobose, ovoid to ellipsoid, or irregular cells, 10–20 × 7–17 μm. *Basidia* (Figure 9b) 35–44 × 9–11 μm, clavate, four-spored; sterigmata up to 4–6 μm long; basal septa lacking clamps. *Basidiospores* [52/2/2] (7.5–) 8.0–10.0 (–11.0) × 6.0–7.5 (–8.0) μm, avl × avw = 9.0 × 6.8 μm, Q = 1.20–1.50 μm, Qm = 1.33 ± 0.09, broadly ellipsoid to ellipsoid, thin-walled, smooth, colorless, amyloid (Figure 9a). *Lamellar edge* sterile, consisting of subglobose to ellipsoid inflated cells (12–37 × 9–25 μm), single or in chains of two–three, thin-walled, colorless; filamentous hyphae fairly abundant, 3–5 μm wide, irregularly arranged. *Pileipellis* 65–90 μm thick, two-layered; upper layer (30–60 μm thick) strongly gelatinized, consisting of radial, thinwalled, colorless, filamentous hyphae 2–4 μm wide; lower layer (30–35 μm thick) consisting of radially and compactly arranged, filamentous hyphae 2–6 μm wide, colorless; vascular hyphae scarce. *Volval remnants* on pileus (Figure 9c) consisting of irregularly arranged elements: inflated cell (23–48 × 20–35 μm), fairly abundant, subglobose to ellipsoid, single or in chains of two–three, colorless; filamentous hyphae (3–10 μm wide) abundant to dominate, colorless or light yellow; vascular hyphae scarce. Inner part of *volval remnants* on stipe base consisting of longitudinally arranged elements: filamentous hyphae very abundant to predominant, 1–8 μm wide, colorless, thin-walled; inflated cells fairly abundant to abundant, subglobose to ovoid, or ellipsoid, 23–55 × 18–37 μm, colorless, thin-walled; vascular hyphae scarce. The outer surface of volval remnants on stipe base similar to structure of interior part but with more abundant inflated cells. Inner surface gelatinized, similar to structure of interior part but with a few inflated cells. *Stipe trama* consists of longitudinally arranged, long clavate terminal cells, 170–310 × 16–27 μm; filamentous hyphae abundant, 3–10 μm wide; vascular hyphae scarce. *Annulus* consisting of loosely arranged, interwoven elements: inflated cells abundant, pyriform to subglobose, 16–43 × 12–24 μm, single or in chains of two–three, colorless, thin-walled; filamentous hyphae scarce to fairly abundant, 2–8 μm wide, colorless, thin-walled; vascular hyphae scarce. *Clamps* absent in all parts of basidioma.

Habitat: Solitary to scattered on soil in tropical deciduous forests dominated by *Dipterocarpus* and *Shorea* species. Basidiomata occurs in the rainy season.

Distribution: This species is currently known in China [2,18,19,67], India [68], Japan [61], Korea [20], and Thailand ([26], this study).

Specimens examined: THAILAND, Chiang Mai Province, Mueang Chiang Mai District, alt. 1143 m, 3 August 2019, Yuan S. Liu, STO-2019-359 (SDBR-STO19-359); Yuan S. Liu, STO-2019-372 (SDBR-STO19-372).

Notes: *Amanita oberwinkleriana* was firstly reported from Japan [61] and has also been found in other Asian countries, e.g., China, India, Korea, and Thailand [2,18,19,20,26,67,68]. It is characterized by small- to medium-sized basidiomata, a smooth and white pileus often tinged yellowish in the center, non-appendiculate pileal margin, attenuate lamellulae, fusiform to napiform stipe base surrounded by limbate volval remnants, as well as the membranous annulus. Our Thai materials are consistent with all the above features.

Morphologically, *Amanita oberwinkleriana* can be easily confused with a number of species, having a white and smooth pileus, limbate volval remnants on the stipe base, and a membranous annulus, e.g., *A. exitialis*, *A. rimosa,* and *A. virosa*. However, *A. exitialis* distinctly differs from the newly described species by having two-spored basidia and much larger basidiospores (9.5–12.0 × 9.0–11.5 μm, Qm = 1.08 ± 0.04) [2,18,19,20,67]. *Amanita rimosa* can be distinguished from *A*. *oberwinkleriana* by having fissured pileal margin, as well as smaller and globose to subglobose basidiospores (7.0–8.5 × 6.5–8.0 μm, Qm = 1.08 ± 0.05) [2,19,55]. *Amanita virosa* differs by having obvious and concolorous squamules on its stipe, as well as much wider and globose to subglobose basidiospores (8.0–11.0 × 8.0–10.0 μm, Qm = 1.07 ± 0.05) [2,19].

Phylogenetically, *Amanita oberwinkleriana* is related to *A. rubiginosa* Q. Cai, Y.Y. Cui and Zhu L. Yang, *A. avellaneifolia* Zhu L. Yang, Y.Y. Cui and Q. Cai, and *A. modesta*. However, this species distinctly differs from the latter three taxa by its small- to medium-sized basidiomata, as well as a smooth and white pileus [2,15,19].

***Amanita* sect. *Validae*** (Fr.) Quél., Mém. Soc. Emul. Montbéliard, Ser. II, 5: 69 (1872).

Basionym: *Agaricus* sect. *Validae* Fr., Monogr. Amanit. Sueciea: 10 (1854).

Lectotype: *Amanita excelsa* (Fr.) Bertill., Dictionnaire encyclopédique des sciences médicales 1 (3): 499 (1866). 

Notes: Previously, six taxa belonging to *Amanita* sect. *Validae* have been reported from Thailand, namely *A. castanea* Thongbai, Tulloss, Raspé and K.D. Hyde, *Amanita* cf. *spissacea* S. Imai, *A. flavipes* S. Imai sensu lato, *A. fritillaria* (Berk.) Sacc., *A. sculpta* Corner and Bas, and *A. sinocitrina* Zhu L. Yang, Zuo H. Chen and Z.G. Zhang [25,26,29]. In this study, *A. cacaina* L.P. Tang, T. Huang and N.K. Zeng and *A. citrinoannulata* Y.Y. Cui, Q. Cai and Zhu L. Yang are recognized and reported as two new records in Thailand on the basis of the phylogenetic and morphological analyses.

***Amanita cacaina*** L.P. Tang, T. Huang and N.K. Zeng, Frontiers Microbiol. 13: 3 (2023). Figure 2k and Figure 10.

*Basidiomata* is very large. *Pileus* 18.0–18.6 cm diam., plano-convex to applanate, brownish orange (7C3–6) to brown (7E5–7); volval remnants on pileus often pyramidal to verrucose, 2–9 mm high and 2–8 mm wide, sometimes scaly, yellowish white (4A2) to dark brown (7F7–8); margin non-striate, appendiculate; context 5–17.0 mm wide, yellowish white (4A2) to light brown (7D4–5). *Lamellae* free, crowded, reddish brown (8E7–8) to dark brown (8F7–8); lamellulae attenuate. *Stipe* 23.8–27.3 cm long × 2.0–2.5 cm diam. (the length includes the basal bulb), cylindrical, light brown (7D4–6), densely covered by floccose to pulverulent white (1A1) to reddish brown (8E7–8) squamules; context solid; basal part 5.3–5.6 cm diam., globose to subglobose, upper part covered with verrucose to squarish, reddish brown (8D5–6) warts. *Annulus* fragile and fugacious. *Odor* not recorded.

*Lamellar trama* bilateral. Mediostratum 20–35 μm wide, consisting of abundant ellipsoid or clavate to fusiform inflated cells (56–135 × 17–35 μm); filamentous hyphae abundant, 2–6 μm wide; vascular hyphae scarce. Lateral stratum 20–30 μm wide, consisting of abundant ellipsoid to fusiform inflated cells (48–73 × 13–23 μm), diverging at an angle of about 45° to the mediostratum; filamentous hyphae abundant, 3–6 μm wide. *Subhymenium* 25–40 μm thick, with two–three layers of subglobose, or irregular cells, 15–28 × 10–18 μm. *Basidia* (Figure 10b) 38–56 × 12–20 μm, clavate, four-spored; sterigmata up to 3–6 μm long; basal septa lacking clamps. *Basidiospores* [50/2/2] (8.0–) 8.5–10.5 (–11.0) × 8.0–10.0 μm, avl × avw = 9.2 × 8.9 μm, Q = 1.00–1.13 (–1.24) μm, Qm = 1.04 ± 0.05, globose to subglobose, sometimes broadly ellipsoid, thin-walled, smooth, colorless to pale yellow, amyloid (Figure 10a). *Lamellar edge* sterile, consisting of pyriform to subglobose, or ellipsoid to clavate inflated cells (13–41 × 9–32 μm), single or in chains of two–three, thin-walled, colorless; filamentous hyphae abundant, 2–6 μm wide, irregularly arranged. *Pileipellis* 70–150 μm thick, two-layered, consisting of radial filamentous hyphae (3–12 μm wide); vascular hyphae scarce. *Volval remnants* on pileus (Figure 10c) consisting of vertically arranged elements: inflated cell (26–150 × 22–87 μm), abundant to predominant, subglobose to ellipsoid, single or in chains of two–three, colorless or light brownish yellow; filamentous hyphae (3–9 μm wide) abundant, colorless or light yellow; vascular hyphae scarce. *Volval remnants* on stipe base predominately composed of irregularly arranged light-brownish yellow, pyriform to subglobose, or ellipsoid to fusiform inflated cells (17–72 × 13–67 μm), mixed with abundant filamentous hyphae (3–7 μm wide). *Stipe trama* consisting of longitudinally arranged, long clavate terminal cells, 170–330 × 18–40 μm; filamentous hyphae abundant, 2–9 (–14) μm wide; vascular hyphae scarce. *Clamps* absent in all parts of basidioma.

Habitat: Solitary to scattered on soil in tropical deciduous or coniferous forests dominated by *Dipterocarpus, Shorea*, and *Pinus* species, respectively. Basidiomata occurs in the rainy season. 

Distribution: This species is currently known in China [69], and Thailand [this study].

Specimens examined: THAILAND, Phetchabun Province, Nam Nao District, alt. 887 m, 17 August 2020, Yuan S. Liu, STO-2020-324 (SDBR-STO20-324); Yuan S. Liu, STO-2020-338 (SDBR-STO20-338).

Notes: Our Thai materials possess very large basidiomata, pyramidal to verrucose and dark-brown pileal remnants, non-striate and appendiculate margin, solid stipe densely covered with floccose to pulverulent white to reddish-brown squamules, globose to subglobose stipe base surrounded by verrucose to squarish, reddish-brown warts. All these features are consistent with the original description of *Amanita cacaina*, which was recently published in China in 2023 [69]. Morphologically, *A. cacaina* is similar to a number of taxa, e.g., *A. pseudosculpta* L.P. Tang and T. Huang, *A. sculpta*, and *A. westii* (Murrill) Murrill. Detailed comparisons between *A. cacaina* and the related species above can be found in Huang et al. [69].

***Amanita citrinoannulata*** Y.Y. Cui, Q. Cai and Zhu L. Yang, Fungal Divers. 91: 186 (2018). Figure 2l and Figure 11.

*Basidiomata is* small. *Pileus* 2.4 cm diam., plano-convex, brownish orange (5C4–6) to yellowish brown (5E7–8); volval remnants on pileus floccose-felted, dark yellow (4C8), densely arranged over disk, often washed away by rain; margin non-striate, non-appendiculate; context white (8A1), slowly changing to pinkish white (8A2) after injury. *Lamellae* free, crowded, white (1A1); lamellulae attenuate. *Stipe* 4.2 cm long × 0.4 cm diam. (the length includes the basal bulb), subcylindrical or slightly tapering upwards, yellowish white (4A2), turning to pinkish white or pale red (8A2–3) when bruised, covered by yellowish-white (4A2) to brown (5E8), snakeskin-shaped squamules above the annulus and fibrous squamules under the annulus; context solid, slowly changing to pinkish white (8A2) after injury; basal part 0.8 cm diam., ventricose, upper part covered with irregular, floccose, yellowish-white (4A2) volval remnants. *Annulus* subapical, membranous, fugacious, yellowish white to pale yellow (3A2–3). *Odor* not recorded.

*Lamellar trama* bilateral. Mediostratum 20–30 μm wide, consisting of abundant ellipsoid or clavate inflated cells (40–85 × 14–28 μm); filamentous hyphae abundant, 2–6 μm wide; vascular hyphae scarce. Lateral stratum 25–40 μm wide, consisting of abundant ellipsoid to clavate or ovoid inflated cells (20–80 × 14–30 μm), diverging at an angle of about 45° to the mediostratum; filamentous hyphae abundant, 2–11 μm wide. *Subhymenium* 20–25 μm thick, with two–three layers of subglobose, or irregular cells, 7–23 × 6–15 μm. *Basidia* (Figure 11b) 30–40 × 8–12 μm, clavate, four-spored; sterigmata up to 3–4 μm long; basal septa lacking clamps. *Basidiospores* [51/1/1] (7.0–) 7.5–9.0 × 6.0–7.5 μm, avl × avw = 8.1 × 6.5 μm, Q = (1.07–)1.14–1.38 (–1.50) μm, Qm = 1.25 ± 0.09, subglobose to broadly ellipsoid, thin-walled, smooth, colorless to light brown, amyloid (Figure 11a). *Lamellar edge* sterile, consisting of globose to subglobose, or ellipsoid inflated cells (18–36 × 17–28 μm), single or in chains of two–three, thin-walled, colorless; filamentous hyphae abundant, 2–6 μm wide, irregularly arranged. *Pileipellis* 100–150 μm thick, two-layered; upper layer (40–70 μm thick) slightly gelatinized, consisting of subradially arranged, thin-walled, colorless to pale yellow, filamentous hyphae 2–10 μm wide; lower layer (60–80 μm thick) consisting of radially and compactly arranged, colorless to pale-yellow filamentous hyphae 2–11 μm wide; vascular hyphae fairly abundant. *Volval remnants* on pileus (Figure 11c) consisting of almost vertically arranged elements: inflated cell (17–52 × 15–40 μm), abundant to predominant, subglobose to globose, single or in chains of two–three, colorless or light brown; filamentous hyphae (2–6 μm wide) scattered, colorless or light yellow; vascular hyphae scarce. *Volval remnants* on stipe base similar to the remnants on pileus but with more filamentous hyphae. *Stipe trama* consists of longitudinally arranged, long clavate to fusiform terminal cells, 63–238 × 20–38 μm; filamentous hyphae abundant, 2–7 μm wide; vascular hyphae scarce. *Annulus* consisting of radially arranged elements: inflated cells scattered, subglobose to clavate or ellipsoid, 20–43 × 12–34 μm, colorless to light brown, thin-walled; filamentous hyphae very abundant to predominant, 2–6 μm wide, colorless, thin-walled; vascular hyphae scarce. *Clamps* absent in all parts of basidioma.

Habitat: Solitary to scattered on soil in tropical deciduous forests dominated by *Dipterocarpus* and *Shorea* species. Basidiomata occurs in the rainy season.

Distribution: This species is currently known in China [2], and Thailand [this study].

Specimens examined: THAILAND, Chiang Mai Province, Mae Taeng District, alt. 720 m, 9 August 2019, Yuan S. Liu, STO-2019-483 (SDBR-STO19-483).

Notes: *Amanita citrinoannulata* was reported from China [2]. Our Thai material has a small basidioma, brownish pileus covered with floccose-felted, with dark-yellow remnants; context of pileus changing to pinkish tone after injury; yellowish stipe always changing to pinkish white or pale red when bruised; snakeskin-shaped squamules above the annulus and fibrous squamules under the annulus. All these features are consistent with the type specimen.

Morphologically, *Amanita citrinoannulata* is similar to a number of taxa, e.g., *A. flavorubens* (Berk. and Mont.) Sacc., *A. fritillaria* Sacc. f. *fritillaria*, and *A. spissacea*. Detailed comparisons between *A. citrinoannulata* and the species mentioned above can be found in [2,19].

Phylogenetically, *Amanita citrinoannulata* is closely related to *A. detersa* Zhu L. Yang, Y.Y. Cui and Q. Cai, *A. flavoconia* G. F. Atk, and *A. spissa* (Fr.) P. Kumm. However, *A. detersa*, reported from China [2], has a grayish-toned pileus, yellowish volva remnants on both pileus and stipe base, and pileal context not changing color when injured. *Amanita flavoconia,* reported from America [70], possesses a pileus that is usually umbonate, the remnants on its pileus are absent or present as yellow patches or powder, and has much narrower basidiospores (6.0–9.0 × 4.0–6.0 μm) [70]. *Amanita spissa* differs in having a robust basidiomata, lacking a color change in the pileal context, the snakeskin-liked squamules on the stipe being absent, and a white annulus [2,19].

## 4. Discussion

In our multigene phylogenetic analysis, the species in each section of the *Amanita* subgen. *Amanitina* are clearly recognized, and the topological structure among these six sections is also clearly presented. In particular, the relationship of the sect. *Phalloideae* and the other three sections, viz. sect. *Arenariae*, sect. *Strobiliformes*, and sect. *Validae,* are clearly recognized. Meanwhile, the species in sect. *Phalloideae* are separated into three subclades, which is consistent with previous studies [2,4,26,54,71,72].

In addition, the *Amanita ballerina*–*A*. *chuformis* subclade and *A. hesleri*–*A*. *zangii* subclade are composed of non-lethal species, which differ morphologically from lethal species from the *A. millsii*–*A*. *virosa* subclade by having elongated to ventricose basal bulbs, striate and appendiculate margins, etc. [15,16,26,54]. Thus, it seems reasonable that two non-lethal subclades could be considered as one or two new sections [4,26]. On the other hand, the *A. millsii*–*A*. *virosa* subclade is composed of lethal species, which often contain several deadly *Amanita* cyclic peptides, e.g., *α*-amanitin, *β*-amanitin, phalloidin, and phallacidin [38,56,73,74,75]. Our two samples and one undescribed specimen (ECM3-PKA) isolated from *Dipterocarpus alatus* roots, representing *A. albifragilis*, nest into the *A. millsii*–*A*. *virosa* subclade and cluster together with two deadly species, viz. *A. griseorosea* and *A. molliuscula*. According to previous studies [38,56], *A. griseorosea* contains *β*-amanitin, and *A. molliuscula* contains both *α*-amanitin and *β*-amanitin. Thus, *A. albifragilis* probably contains both compounds or at least one of them, and this speculation will be clarified in our next project. 

## Figures and Tables

**Figure 1 jof-09-00601-f001:**
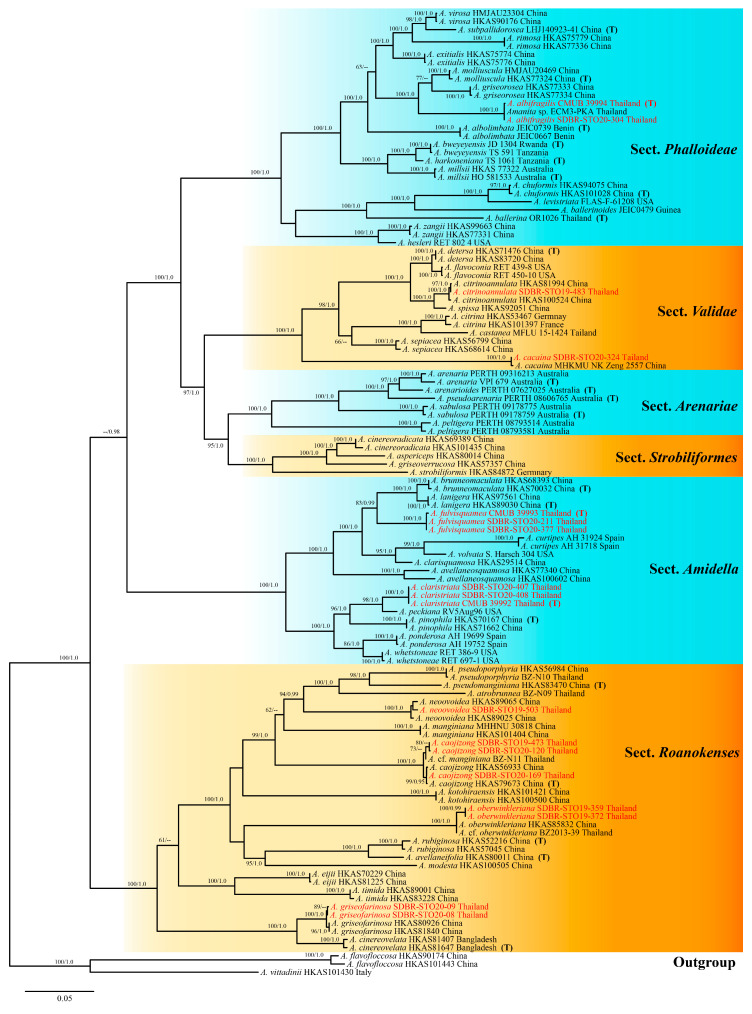
RAxML tree based on a concatenated dataset (nrLSU + ITS + *RPB2* + *TEF1-a* + *TUB*). Bootstrap values (BS) for ML ≥ 60% and posterior probabilities (PPs) for BI ≥ 0.95 are placed above or below the branches. Newly generated sequences are indicated in red, and sequences from type material are marked with (T). The tree is rooted with *Amanita flavofloccosa* (HKAS101443 and HKAS90174) and *A. vittadinii* (HKAS101430).

**Figure 2 jof-09-00601-f002:**
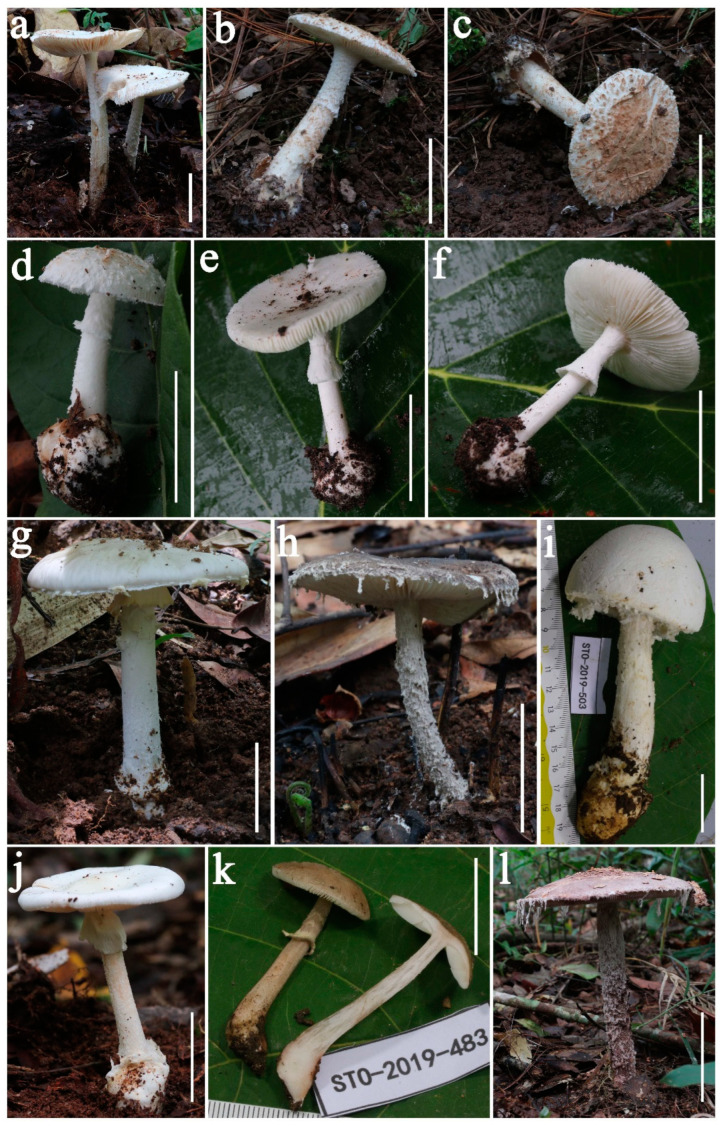
Fresh basidiomata of studied *Amanita* species. (**a**) *A. claristriata* (CMUB39992, holotype). (**b**–**d**) *A. fulvisquamea* [(**b**,**c**) CMUB39993, holotype; (**d**) SDBR-STO20-211)]. (**e**,**f**) *A. albifragilis* (CMUB39994, holotype). (**g**) *A. caojizong* (SDBR-STO20-120). (**h**) *A. griseofarinosa* (SDBR-STO20-08). (**i**) *A. neoovoidea* (SDBR-STO19-503). (**j**) *A. oberwinkleriana* (SDBR-STO19-359). (**k**) *A. citrinoannulata* (SDBR-STO19-483). (**l**) *A. cacaina* (SDBR-STO20-324). Scale bars: (**a**–**k**) = 3 cm, (**l**) = 9 cm.

**Figure 3 jof-09-00601-f003:**
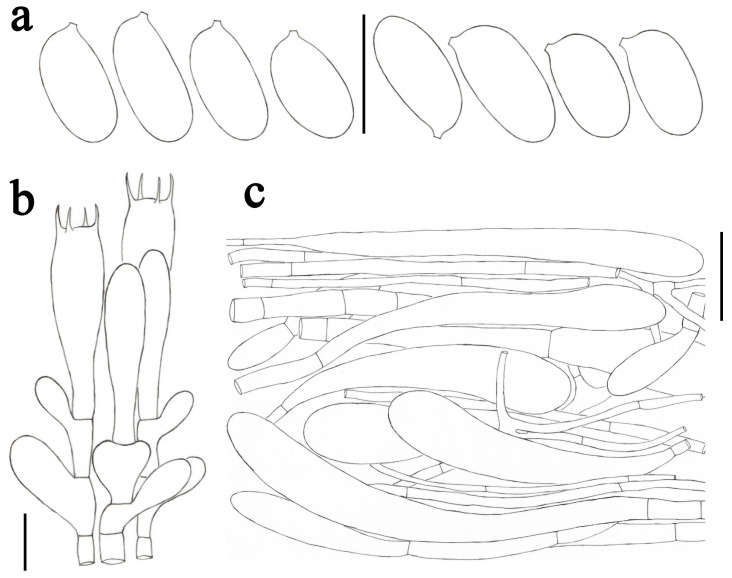
*Amanita claristriata* (CMUB39992, holotype). (**a**) Basidiospores. (**b**) Hymenium and subhymenium. (**c**) Longitudinal section of volval remnants on pileus. Scale bars: (**a**,**b**) = 10 μm, (**c**) = 50 μm.

**Figure 4 jof-09-00601-f004:**
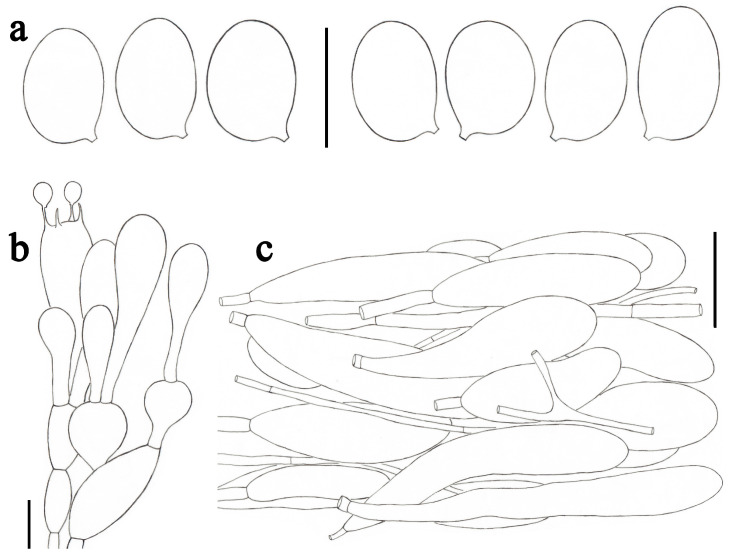
*Amanita fulvisquamea* (CMUB39993, holotype). (**a**) Basidiospores. (**b**) Hymenium and subhymenium. (**c**) Longitudinal section of volval remnants on pileus. Scale bars: (**a**,**b**) = 10 μm, (**c**) = 50 μm.

**Figure 5 jof-09-00601-f005:**
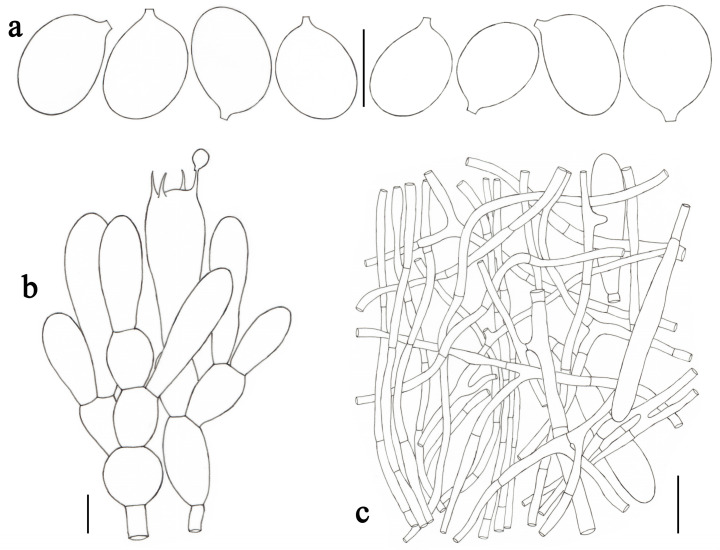
*Amanita albifragilis* (CMUB39994, holotype). (**a**) Basidiospores. (**b**) Hymenium and subhymenium. (**c**) Longitudinal section of outer surface of volval remnants on stipe base. Scale bars: (**a**,**b**) = 10 μm, (**c**) = 50 μm.

**Figure 6 jof-09-00601-f006:**
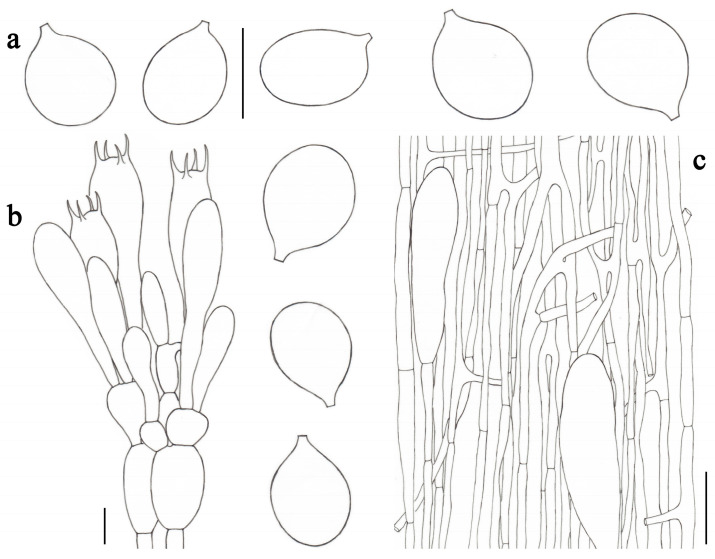
*Amanita caojizong* (SDBR-STO20-120). (**a**) Basidiospores. (**b**) Hymenium and subhymenium. (**c**) Longitudinal section of outer surface of volval remnants on stipe base. Scale bars: (**a**,**b**) = 10 μm, (**c**) = 50 μm.

**Figure 7 jof-09-00601-f007:**
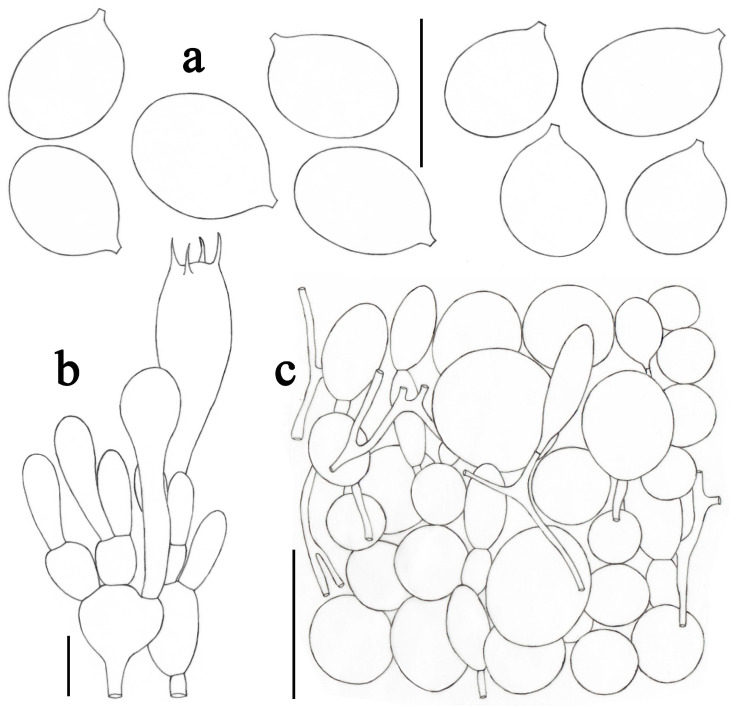
*Amanita griseofarinosa* (SDBR-STO20-09). (**a**) Basidiospores. (**b**) Hymenium and subhymenium. (**c**) Longitudinal section of volval remnants on pileus. Scale bars: (**a**,**b**) = 10 μm, (**c**) = 50 μm.

**Figure 8 jof-09-00601-f008:**
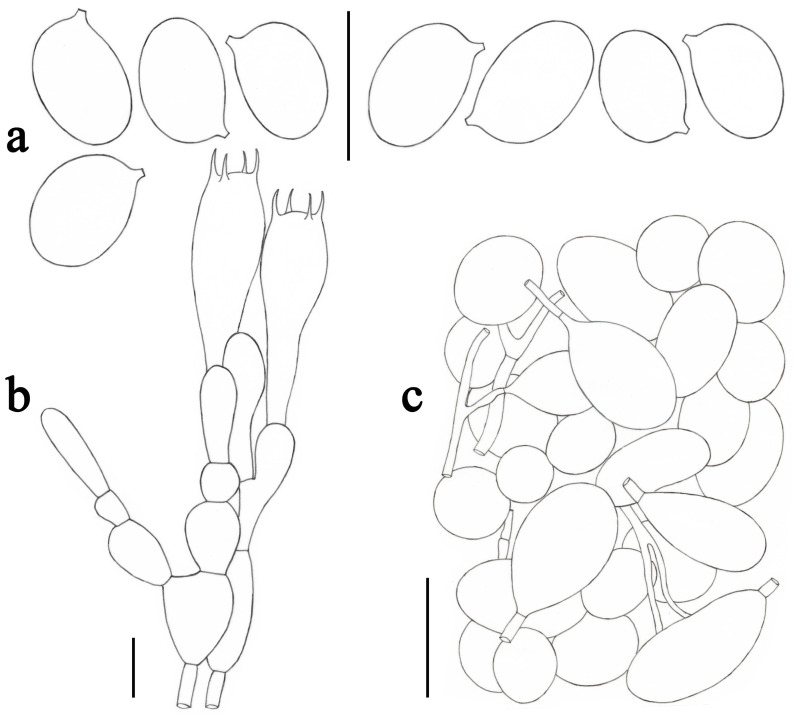
*Amanita neoovoidea* (SDBR-STO19-503). (**a**) Basidiospores. (**b**) Hymenium and subhymenium. (**c**) Longitudinal section of volval remnants on pileus. Scale bars: (**a**,**b**) = 10 μm, (**c**) = 50 μm.

**Figure 9 jof-09-00601-f009:**
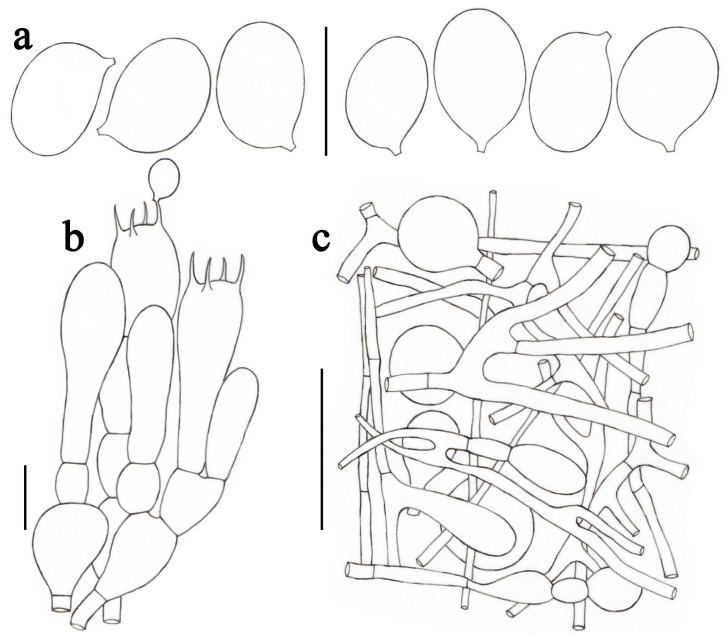
*Amanita oberwinkleriana* (SDBR-STO19-372). (**a**) Basidiospores. (**b**) Hymenium and subhymenium. (**c**) Longitudinal section of volval remnants on pileus. Scale bars: (**a**,**b**) = 10 μm, (**c**) = 50 μm.

**Figure 10 jof-09-00601-f010:**
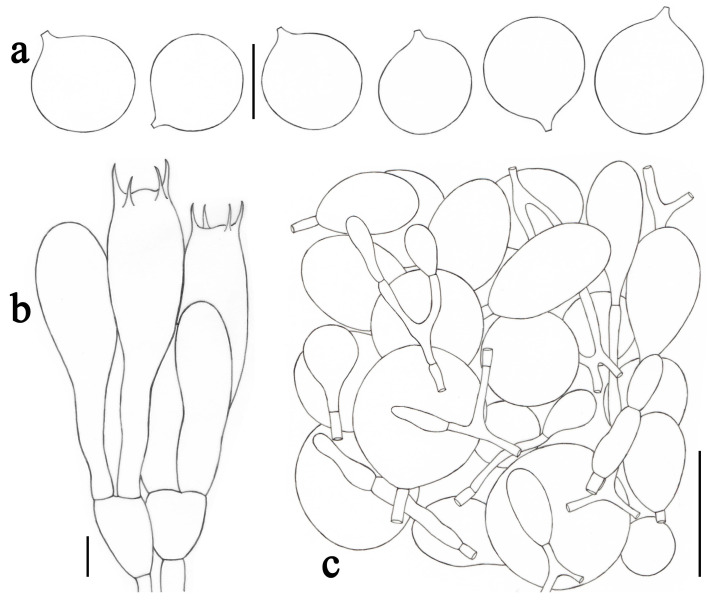
*Amanita cacaina* (SDBR-STO20-324). (**a**) Basidiospores. (**b**) Hymenium and subhymenium. (**c**) Longitudinal section of volval remnants on pileus. Scale bars: (**a**,**b**) = 10 μm, (**c**) = 50 μm.

**Figure 11 jof-09-00601-f011:**
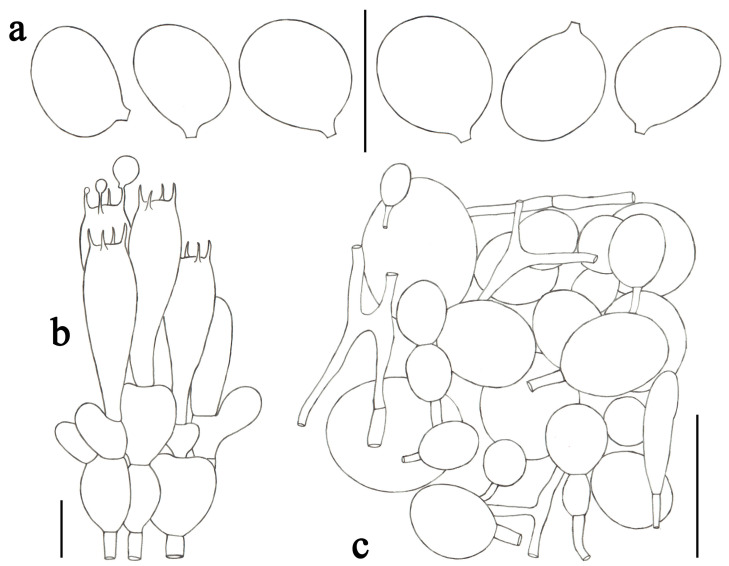
*Amanita citrinoannulata* (SDBR-STO19-483). (**a**) Basidiospores. (**b**) Hymenium and subhymenium. (**c**) Longitudinal section of volval remnants on pileus. Scale bars: (**a**,**b**) = 10 μm, (**c**) = 50 μm.

**Table 1 jof-09-00601-t001:** Species names, voucher numbers, countries, and GenBank accession numbers of the taxa used in this study.

Species Names	Voucher Numbers	Countries	GenBank Accession Numbers				
			nrLSU	ITS	*RPB2*	*TEF1-α*	*TUB*
Section *Amidella*							
*A. avellaneosquamosa*	HKAS77340	China	KJ466483	KJ466418	KJ466648	KJ481982	KJ466562
*A. avellaneosquamosa*	HKAS100602	China	MH486379	MH508258	MH485873	MH508681	—
*A. brunneomaculata*	HKAS68393	China	MH486410	MH508278	MH485892	MH508698	MH485428
*A. brunneomaculata*	HKAS70032 ^T^	China	MH486411	MH508279	MH485893	MH508699	—
*A. clarisquamosa*	HKAS29514	China	AF024448	—	—	—	—
* A. claristriata *	CMUB39992 ^T^	Thailand	OQ780668	OQ780686	OQ740048	OQ740066	—
* A. claristriata *	SDBR-STO20-407	Thailand	OQ780669	OQ780687	OQ740049	OQ740067	OQ740085
* A. claristriata *	SDBR-STO20-408	Thailand	OQ780670	OQ780688	OQ740050	OQ740068	OQ740086
*A. curtipes*	AH 31718	Spain	EF653961	AY486233	—	—	—
*A. curtipes*	AH 31924	Spain	EF653960	EF653963	—	—	—
* A. fulvisquamea *	CMUB39993 ^T^	Thailand	OQ780671	OQ780689	OQ740051	OQ740069	OQ740087
* A. fulvisquamea *	SDBR-STO20-211	Thailand	OQ780672	OQ780690	OQ740052	OQ740070	—
* A. fulvisquamea *	SDBR-STO20-377	Thailand	OQ780673	OQ780691	OQ740053	OQ740071	—
*A. lanigera*	HKAS89030 ^T^	China	MH486621	MH508420	MH486074	MH508880	—
*A. lanigera*	HKAS97561	China	MH486622	MH508421	MH486075	—	MH485591
*A. peckiana*	RV5Aug96	USA	AF097387	—	—	—	—
*A. pinophila*	HKAS70167 ^T^	China	MH486759	MH508504	MH486178	—	MH485682
*A. pinophila*	HKAS71662	China	MH486760	MH508505	MH486179	—	MH485683
*A. ponderosa*	AH 19752	Spain	EF653958	AY486234	—	—	—
*A. ponderosa*	AH 19699	Spain	EF653959	EF653962	—	—	—
*A. volvata*	S. Harsch 304	USA	AF024485	—	—	—	—
*A. whetstoneae*	RET 386-9	USA	KX061533	KX061519	—	—	—
*A. whetstoneae*	RET 697-1	USA	KX061531	KX061518	—	—	—
Section *Arenariae*							
*A. arenaria*	VPI 679 ^T^	Australia	GQ925382	GQ925393	—	—	—
*A. arenaria*	PERTH 09316213	Australia	MW793397	MW795714	MW820674	MW820649	MW820671
*A. arenarioides*	PERTH 07627025 ^T^	Australia	MW775283	MW775309	MW820678	MW820655	MW820664
*A. peltigera*	PERTH 08793514	Australia	MN900625	MN894307	MN912054	MN909824	—
*A. peltigera*	PERTH 08793581	Australia	MN900627	MN894321	MN912056	MN909826	MN905762
*A. pseudoarenaria*	PERTH 08606765 ^T^	Australia	MW775284	MW775312	MW820681	MW820656	MW820669
*A. sabulosa*	PERTH 09178759 ^T^	Australia	MW775279	MW775291	MW820675	MW820650	MW820660
*A. sabulosa*	PERTH 09178775	Australia	MW775280	MW775296	MW820677	MW820653	MW820661
Section *Phalloideae*							
* A. albifragilis *	CMUB39994 ^T^	Thailand	OQ780674	OQ780692	—	OQ740072	—
* A. albifragilis *	SDBR-STO20-304	Thailand	OQ780675	OQ780693	—	OQ740073	OQ740088
*A. albolimbata*	JEIC0667	Benin	MT966939	MT966932	MT966958	MT966953	MT966947
*A. albolimbata*	JEIC0739 ^T^	Benin	MT966942	MT966935	MT966963	MT966955	MT966950
*A. ballerina*	OR1026 ^T^	Thailand	NG_058607	KY747467	KY656884	—	KY656865
*A. ballerinoides*	JEIC0479	Guinea	OK510856	OK510854	OK510847	OK510824	OK510835
*A. bweyeyensis*	JD 1304 ^T^	Rwanda	MK570927	MK570920	MK570937	MK570940	MK570916
*A. bweyeyensis*	TS 591	Tanzania	MK570928	MK570921	—	—	—
*A. chuformis*	HKAS94075	China	MT395380	MT395378	—	MT364256	—
*A. chuformis*	HKAS101028 ^T^	China	MT395381	MT395379	MT364258	MT364257	—
*A. exitialis*	HKAS75774	China	JX998052	JX998027	KJ466591	JX998001	KJ466503
*A. exitialis*	HKAS75776	China	JX998051	JX998025	KJ466593	JX998003	KJ466505
*A. griseorosea*	HKAS77333	China	KJ466475	KJ466412	KJ466660	—	KJ466579
*A. griseorosea*	HKAS77334	China	KJ466476	KJ466413	KJ466661	—	KJ466580
*A. harkoneniana*	TS 1061 ^T^	Tanzania	MK570930	MK570923	—	—	—
*A. hesleri*	RET 802-4	USA	MH836560	MH836568	—	—	—
*A. levistriata*	FLAS-F-61208	USA	MH620278	MH211813	—	—	—
*A. millsii*	HO 581533 ^T^	Australia	KY977713	KY977714	MF000753	MF000759	MF000760
*A. millsii*	HKAS 77322	Australia	KJ466457	KJ466395	KJ466643	KJ481978	KJ466557
*A. molliuscula*	HKAS77324 ^T^	China	KJ466472	KJ466409	KJ466639	KJ481974	KJ466553
*A. molliuscula*	HMJAU20469	China	KJ466473	KJ466410	KJ466640	KJ481975	KJ466554
*A. rimosa*	HKAS75779	China	JX998046	JX998020	KJ466617	JX998004	KJ466528
*A. rimosa*	HKAS77336	China	KJ466456	KJ466394	KJ466622	KJ481958	KJ466533
*Amanita* sp.	ECM3-PKA	Thailand	—	DQ146367	—	—	—
*A. subpallidorosea*	LHJ140923-41 ^T^	China	KP691692	KP691683	KP691701	KP691670	KP691711
*A. virosa*	HKAS90176	China	MH486948	MH508650	MH486341	MH509167	MH485847
*A. virosa*	HMJAU23304	China	KJ466498	KJ466431	KJ466667	KJ481999	KJ466587
*A. zangii*	HKAS99663	China	MH486958	MH508655	MH486351	MH509178	MH485855
*A. zangii*	HKAS77331	China	KJ466500	KJ466433	KJ466669	KJ482001	KJ466589
Section *Roanokenses*							
*A. atrobrunnea*	BZ-N09	Thailand	KT934314	KY747455	KY656871	—	KY656852
*A. avellaneifolia*	HKAS80011 ^T^	China	MH486378	—	MH485872	MH508680	MH485410
*A. caojizong*	HKAS56933	China	KJ466438	KJ466378	KJ466603	KJ481943	KJ466515
*A. caojizong*	HKAS79673 ^T^	China	MH486429	MH508291	MH485909	MH508714	—
* A. caojizong *	SDBR-STO19-473	Thailand	OQ780676	—	OQ740054	OQ740074	OQ740089
* A. caojizong *	SDBR-STO20-120	Thailand	OQ780677	—	OQ740055	OQ740075	OQ740090
* A. caojizong *	SDBR-STO20-169	Thailand	OQ780678	—	OQ740056	OQ740076	—
*A.* cf. *manginiana*	BZ-N11	Thailand	KY747474	KY747457	KY656873	—	KY656854
*A.* cf. *oberwinkleriana*	BZ2013-39	Thailand	KY747476	KY747459	KY656876	—	KY656857
*A. cinereovelata*	HKAS81647 ^T^	Bangladesh	KP259291	—	KP259288	KP259289	—
*A. cinereovelata*	HKAS 81407	Bangladesh	KP259292	—	—	KP259290	—
*A. eijii*	HKAS70229	China	MH486484	MH508333	MH485963	MH508761	MH485486
*A. eijii*	HKAS81225	China	MH486485	—	MH485964	MH508762	MH485487
*A. griseofarinosa*	HKAS80926	China	MH486559	MH508375	MH486025	MH508830	MH485545
*A. griseofarinosa*	HKAS81840	China	MH486560	—	MH486026	MH508831	MH485546
* A. griseofarinosa *	SDBR-STO20-08	Thailand	OQ780679	—	OQ740057	OQ740077	OQ740091
* A. griseofarinosa *	SDBR-STO20-09	Thailand	OQ780680	—	OQ740058	OQ740078	OQ740092
*A. kotohiraensis*	HKAS100500	China	MH486613	MH508414	—	—	—
*A. kotohiraensis*	HKAS101421	China	MH486615	—	MH486069	MH508875	MH485587
*A. manginiana*	HKAS101404	China	MH486637	—	—	—	—
*A. manginiana*	MHHNU 30818	China	MH605436	—	MH614263	MH614264	MH614265
*A. modesta*	HKAS100505	China	MH486647	MH508438	MH486098	MH508904	MH485613
*A. neoovoidea*	HKAS89025	China	MH486656	MH508445	MH486106	MH508913	MH485621
*A. neoovoidea*	HKAS89065	China	MH486657	—	MH486107	MH508914	MH485622
* A. neoovoidea *	SDBR-STO19-503	Thailand	—	—	OQ740060	OQ740080	—
*A. oberwinkleriana*	HKAS85832	China	MH486681	—	MH486118	—	MH485632
* A. oberwinkleriana *	SDBR-STO19-359	Thailand	OQ780682	OQ780694	OQ740061	OQ740081	—
* A. oberwinkleriana *	SDBR-STO19-372	Thailand	OQ780683	OQ780695	OQ740062	OQ740082	—
*A. pseudomanginiana*	HKAS83470 ^T^	China	MH486772	—	MH486187	—	MH485694
*A. pseudoporphyria*	BZ-N10	Thailand	KY747473	KY747456	KY656872	—	KY656853
*A. pseudoporphyria*	HKAS56984	China	KJ466450	KC429050	KJ466614	KJ481953	KJ466525
*A. rubiginosa*	HKAS52216 ^T^	China	MH486817	MH508561	MH486229	—	MH485734
*A. rubiginosa*	HKAS57045	China	MH486819	MH508563	MH486231	—	MH485736
*A. timida*	HKAS83228	China	MH486930	MH508636	MH486323	MH509147	MH485830
*A. timida*	HKAS89001	China	MH486932	—	MH486325	MH509149	—
Section *Strobiliformes*							
*A. aspericeps*	HKAS80014	China	MH486374	MH508257	MH485868	MH508676	MH485408
*A. cinereoradicata*	HKAS101435	China	MH486451	MH508307	MH485933	MH508729	MH485458
*A. cinereoradicata*	HKAS69389	China	MH486453	MH508308	MH485934	MH508730	MH485459
*A. griseoverrucosa*	HKAS57357	China	MH486581	MH508392	MH486043	MH508850	MH485562
*A. strobiliformis*	HKAS84872	Germany	MH486895	MH508614	MH486298	MH509117	MH485798
Section *Validae*							
*A. cacaina*	MHKMU NK Zeng 2557	China	ON768725	ON768705	—	—	—
* A. cacaina *	SDBR-STO20-324	Thailand	OQ780684	OQ780696	OQ740063	OQ740083	—
*A. castanea*	MFLU 15-1424	Thailand	KU877539	KU904823	—	—	—
*A. citrina*	HKAS53467	Germany	MH486457	MH508312	MH485937	MH508733	MH485461
*A. citrina*	HKAS101397	France	MH486456	MH508311	MH485936	MH508732	MH485460
*A. citrinoannulata*	HKAS81994	China	MH486463	MH508317	MH485943	MH508739	MH485466
*A. citrinoannulata*	HKAS100524	China	MH486459	MH508314	MH485939	MH508735	MH485463
* A. citrinoannulata *	SDBR-STO19-483	Thailand	OQ780685	OQ780697	OQ740065	OQ740084	—
*A. detersa*	HKAS71476 ^T^	China	MH486475	MH508328	MH485954	MH508752	MH485479
*A. detersa*	HKAS83720	China	MH486479	MH508332	MH485958	MH508756	MH485482
*A. flavoconia*	RET 439-8	USA	MH486511	MH508348	MH485983	MH508787	—
*A. flavoconia*	RET 450-10	USA	MH486512	MH508349	MH485984	MH508788	—
*A. sepiacea*	HKAS56799	China	MH486847	MH508584	MH486256	—	MH485759
*A. sepiacea*	HKAS68614	China	MH486851	MH508585	MH486260	—	MH485763
*A. spissa*	HKAS92051	China	MH486892	MH508611	MH486295	MH509114	MH485795
Outgroup							
*A. flavofloccosa*	HKAS90174	China	KT833801	MH508352	KT833818	KT833831	MH485508
*A. flavofloccosa*	HKAS101443	China	MH486515	—	MH485986	MH508791	MH485507
*A. vittadinii*	HKAS101430	Italy	MH486950	MH508651	MH486342	MH509169	—

Newly generated sequences in this study are in red. Holotypes are marked with “^T^”.

## Data Availability

The DNA sequence data obtained from this study were deposited in GenBank under accession numbers: ITS (OQ780686–OQ780697); nrLSU (OQ780668–OQ780685); *RPB2* (OQ740048–OQ740065); *TEF1-α* (OQ740066–OQ740084); *TUB* (OQ740085–OQ740093).

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
