# Peer review of "Taxonomic Novelties and New Records of *Amanita* Subgenus *Amanitina* from Thailand"

_jof, 2023, doi:10.3390/jof9060601_

Round 1

Reviewer 1 Report

Dear authors,

It was my plaesure to review manuscript entitled "Taxonomic Novelties and New Records of Amanita Subgenus Amanitina from Thailand". I hope that following my suggestions would be good for improving the overall quality of the manuscript submitted.

The authors studied morphology and molecular characters of 20 dried fungal samples belonging to nine species of Amanita subgen. Amanitina from Thailand. Out of these, three species are described as new to science in the manuscript and four species are for the first time recorded in Thailand. Multi-gene phylogenetic analyses including five gene markers (ITS, nrLSU, RPB2, TEF1-α, and TUB) of the genus Amanita are presented and all new species are well supported. In total, 71 gene sequences are newly generated in the study. Morphological characters of all species are described in detail and supplemented with line drawings, basidiomata photographs and comparied with related taxa.

Authors followed the newest International code of nomenclature for algae, fungi, and plants. Descriptions and photographs/line drawings cover all elements needed.

Major points:  (1) Quality of English language is not good.

(2) At least for holotype collections geographic coordinates should be added.

All suggested corrections/additions are included in the attached review of the manuscript file (pdf).

Best, Reviewer

The English language used in the manuscript needs to be thoroughly revised due to many grammatical errors. Some parts of the text are not easily understandable. I tried my best to polish the English but it needs to be reworked by some native speaker also.

Author Response

We really appreciate your efforts on our manuscript (jof-2364091) entitled “Taxonomic Novelties and New Records of Amanita Subgenus Amanitina from Thailand”. We have seriously considered your comments and suggestions to improve the quality of our manuscript.

Here we provided our responses to your comments point by point, please kindly see the details as fellows. Thank you very much.

Sincerely

Yuan S. Liu

Response to reviewers’ comments

Point 1: Quality of English language is not good.

Response: We really appreciate your intensive review. It may take you much time to read and point out the errors.

We do realize that the writing of the previous manuscript was not quite good. Here we attached the revised manuscript which has been edited by the native speaker.

Point 2: At least for holotype collections geographic coordinates should be added. All suggested corrections/additions are included in the attached review of the manuscript file (pdf).

Response: Thank you for your kind suggestions and we have provided the geographic coordinates for three holotypes.

Reviewer 2 Report

This is an interesting work about the species diversity of Amanita subgenus Amanitina from Thailand. The authors employed both morphological and multi-locus phylogenetic evidence to delimit the species and section within the genus Amanita. Three new species ad four species new to Thailand are reported. Detailed description, good line drawing illustrations, and useful comparison with related taxa are provided. However, there are many minor flaws in the manuscript, which have been pointed out or annotated in the manuscript by the reviewer. It is suggested that the authors improve the manuscript carefully.

There are some English flaws in the manuscript. See annotated manuscript.

Author Response

We really appreciate your efforts on our manuscript (jof-2364091) entitled “Taxonomic Novelties and New Records of Amanita Subgenus Amanitina from Thailand”. We have seriously considered your comments and suggestions to improve the quality of our manuscript.

Please see the details in the attachment.

Thank you very much.

Sincerely

Yuan S. Liu

Reviewer 3 Report

This is a very good report, offered by a mature scientific group flauting a remarkable knowledge, and skillfully exploring the described materials to great detail. Unfortunately I could not review all descriptions due to time limitations, but most of what can be seen as amendments and suggestions for change in the few ones that I have revised will, hopefully, guide the Authors through further revisions that need to be done.

I strongly recommend that the Authors elabrate more, in the discussion (and in the abstract), on the significance (taxonomical, chorological, ecological...) of these results. I feel that, while this additional content is not indispensable, it will reinforce the value of this contribution.

Due to the quality of the scientific content, most of my remarks (on the attached file) concern clarity or correctness of language. The Authours should understand that, in some descriptions (still) without amendments/suggestions of my part, most of what is needed will be very similar to the first few ones that I have reviewed.

Author Response

Dear reviewer,

Thank you for your kind suggestions. You have pointed out many errors about English writing. We do realize that the writing of the previous manuscript was not good. Here we attached the revised manuscript which has been edited by the native speaker.

Sincerely

Yuan S. Liu

Round 2

Reviewer 1 Report

Dear authors and the editors,

The manuscript is now improved a lot according to to the most of my instructions. I noticed a few points that could be additionally improved. Please find my remarsks in the pdf version of the manuscript attached. Otherwise the manuscript is fine.

Best wishes, Reviewer

English language is much better in this new version of the text, but could be slightly improved.

Author Response

Dear reviewer,

We really appreciate your professional comments on our manuscript.

We have revised our manuscript following your comments, please kindly see the attachment.

Sincerely

Yuan S. Liu

Reviewer 3 Report

The manuscript is substantially improved in terms of form and is nearly ready for publication. My only objection is on corrections that were not implemented yet. Perhaps the authors forgot to look them up, since they had the assistance of a "native speaker". See example below.

Given this, I advise the authors to check every suggestion (by me and other reviewers) thoroughly so that one can select 'English language fine. No issues detected.' The manuscript is not there yet.

Example of one overlooked correction:

On pages 2-3 (v2), the phrase «Sections of the stipitipellis were longitudinally cut along the small pieces which was taken from the middle part of the stipe» contains a grammatical consistency error ('small pieces... were', not 'was'), 'along' is, to me, a disturbing insertion (with this, I mean that it is superfluous and degrades the clarity of the phrase), and 'which was' is to be deleted.

In my understanding, the authors intend to explain that they made the cuts 'longitudinally, from small pieces taken from the middle part of the stipe', as I suggested that it should be phrased. The word 'longitudinally' is enough to define the plane of sectioning, and 'from' is the adequate adverb.

Perhaps they do not agree with my interpretation, but then their text should be made free of ambiguous meanings.

Author Response

Dear reviewer,

We really appreciate your professional suggestions on our manuscript.

We did follow your comments to revise our manuscript. For the overlooked corrections as you mentioned, we revised the writing following your comments. It is indeed much clearer and better.  Please kindly see the attachment.

Sincerely

Yuan S. Liu
